# One Snapshot, Many Clues: Inverse Protocol Prediction from Single-View Spheroid Images

## Abstract

Understanding how experimental protocols shape spheroid morphology is crucial for advancing 3D cell culture research, yet reconstructing these conditions from imaging alone has remained elusive. We used deep learning frameworks which are able to infer the full experimental protocol including cell line, medium, seeding density, timepoint, formation method, microscope, and magnification from a single bright-field spheroid image. Using the SLiMIA dataset of 8,000 annotated images spanning diverse culture conditions, we cast this as a structured multi-label prediction task and benchmarked a spectrum of models, from CNNs and transformers to hybrid and dependency-aware architectures. Our approach integrates segmentation for morphology extraction, domain-adversarial training, and morphologically informed augmentation to improve robustness across imaging setups. Results show an average accuracy of 95.23% across protocol components, with hybrid models such as CoAtNet excelling in balancing efficiency and accuracy, while feature-augmented and hierarchical models contribute interpretability and consistency. Grad-CAM analyses confirm that predictions rely on biologically meaningful features (e.g., compactness, necrotic core structure), while highlighting dataset-driven artifacts in replicate and magnification tasks.

## 1 Introduction and Background

Three-dimensional (3D) cell culture systems such as spheroids and organoids are central tools in cancer biology, drug discovery, and tissue engineering (Fatehullah et al., 2016; Chatzinikolaidou, 2016). Unlike two-dimensional cultures, spheroids recreate nutrient and oxygen gradients, cell–cell interactions, and necrotic cores, while remaining amenable to bright-field and phase-contrast microscopy for high-throughput, non-destructive readouts (Edmondson et al., 2014). Despite this ubiquity, imaging is still used primarily to measure outcomes (size, viability, morphology), not to infer the experimental protocols that produced them. We pose a new challenge: given a single spheroid image, can we reconstruct the protocol conditions—including cell line, medium, seeding density, timepoint, formation method, and microscope? We term this the inverse protocol prediction problem. Success would enable reproducibility checks, automated experiment validation, and deeper insight into how protocol choices manifest morphologically.

Recent advances in biomedical image segmentation have leveraged deep convolutional neural networks (CNNs) and transformer architectures to handle the complexity of 3D cell culture images. U-Net variants and DeepLab models achieve high segmentation accuracy for spheroid morphology under challenging visual conditions such as noise and imaging artifacts (Park et al., 2023; Liu et al., 2021). Transformer-based architectures like Swin-UNet enhance segmentation by capturing global contextual cues and improving recognition of intricate morphological features (Khan et al., 2023). Semi-supervised learning and multi-label deep supervision have been employed to overcome limitations imposed by scarce annotations and class imbalance, improving model robustness and generalization (Reiß et al., 2021; Han et al., 2024). Synthetic training data generated based on biophysical principles has further augmented data diversity, yielding improved alignment with real-world spheroid structures (Koetzier et al., 2024). In the context of IPP from biomedical images, recent studies have adopted transformer-based and multi-label deep learning frameworks to infer experimental protocols or biological states with high fidelity. Multi-label learning models effectively capture dependencies between protocol components, while transformer architectures provide interpretability and robust feature extraction for heterogeneous biomedical data (Zhang et al.,

2022; Madan et al., 2024). These data-driven methods provide a principled alternative to classical probabilistic IPP, leveraging complex spatial and contextual cues for accurate prediction of underlying experimental conditions. Spatiotemporal modeling for time series prediction in biomedical microscopy has progressed through convolutional LSTM and attention-based deep learning models. Integrating 3D cell culture systems with advanced AI frameworks enables dynamic prediction of morphological progression and treatment response (Dave et al., 2025; Torro et al., 2025). Models such as ConvLSTM and physics-inspired recurrent networks demonstrate strong performance in capturing temporal dependencies in longitudinal microscopy data (De Cillis et al., 2025; Mali et al., 2025). Leveraging metadata and multi-modal inputs further enhances these models' ability to generalize across diverse experimental conditions.

Inverse protocol prediction (IPP) is difficult for several reasons. First, *morphological ambiguity* arises when distinct protocols yield visually similar spheroids. Second, *imaging variability* across microscopes and magnifications introduces artifacts that obscure morphology. Third, few datasets couple high-resolution spheroid images with detailed experimental metadata, limiting model development and evaluation. To address this, we introduce a unified framework for inverse protocol prediction using SLiMIA, a dataset of ∼8,000 bright-field spheroid images spanning nine microscopes, 47 cell lines, multiple media, seeding densities, timepoints, and formation methods (Blondeel et al., 2025). We frame the problem as a structured multi-label prediction, disentangle morphology from imaging artifacts via domain-adversarial training and augmentation, and benchmark convolutional, transformer, and hybrid architectures. While many components are established, our novelty lies in their adaptation to the structured multi-label nature of experimental protocols, where causal label dependencies and morphometric priors are biologically grounded rather than arbitrary.

Our main contributions are summarized as:

1. Morphometry fusion that provides integration of classical shape features (e.g., area, compactness) with deep embeddings.

2. Hierarchical modeling for a multi-task transformer that explicitly captures dependencies among protocol labels.

3. Grad-CAM analysis at protocol-level for interpretability that highlights morphological regions and dataset artifacts.

4. The first temporal modeling of SLiMIA via ConvLSTM, PredRNN++, and PhyDNet to predict the evolution of spheroid (Spatio-temporal extension).

This work demonstrates that spheroid morphology encodes rich, recoverable signatures of culture conditions, establishing microscopy-driven inverse protocol prediction as a new paradigm for reproducibility, optimization, and design in cell culture systems.

## 2 MATERIALS AND METHODS

SLiMIA (Spheroid Light Microscopy Image Atlas) is an open-access morphometric image dataset designed to support machine learning and computational modeling of three-dimensional (3D) cell culture systems. SLiMIA comprises approximately 8,000 light microscopy images of spheroids spanning a diverse range of experimental conditions, including nine microscope types, 47 distinct cell lines, eight culture media, four spheroid formation protocols, and multiple cell seeding densities, accompanied by rich metadata to facilitate reproducible analysis and benchmarking(Blondeel et al., 2025). Figure 1 highlights some sample dataset images along wih their manual segmentatons. Figure 2 demonstrates the entire workflow followed for analysis of the dataset. The same is described in detail in this section.

All segmentation models were trained on RGB inputs with a single binary output mask, where sigmoid activation was applied within the loss or metric functions. We adopted Adam-based optimizers (Adam et al., 2014) and employed a ReduceLROnPlateau scheduler with a patience of 5 and decay factor of 0.5 to adaptively lower the learning rate when validation performance plateaued (Smith, 2017). The choice of adaptive optimizers ensured stable convergence across diverse architectures, while dynamic scheduling helped prevent overfitting. Loss functions were selected to balance pixel-wise accuracy and region overlap, with particular attention to the class imbalance and faint boundaries common in spheroid images. U-Net++ and DeepLabV3+ models employed the Focal Tversky

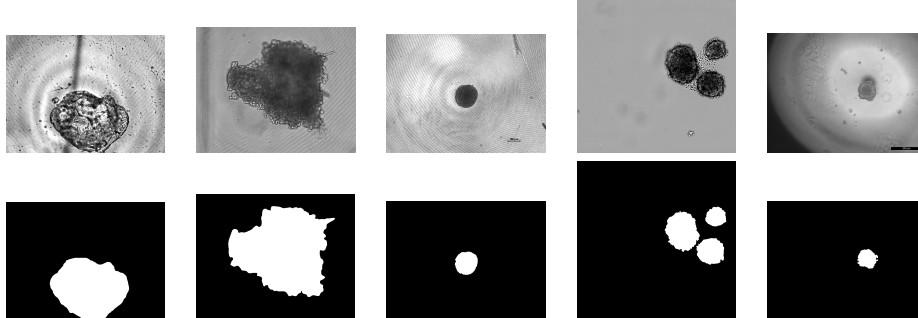

Figure 1: Sample images from the SLiMIA dataset with their manual segmentations (bottom row).

Loss (Salehi et al., 2017) ($\alpha = 0.7, \beta = 0.3, \gamma = 0.75$), chosen for its ability to emphasize recall and mitigate false negatives in imbalanced biomedical data. U-Net++ used a ResNet-50 encoder pretrained on ImageNet, following (Zhou et al., 2018), while DeepLabV3+ also leveraged a ResNet-50 backbone to provide stronger representational capacity compared to shallower variants (Chen et al., 2018a). DeepLabV3 and Attention U-Net were trained with a hybrid loss combining BCEWith-Logits and Dice (0.5 each), encouraging overlap accuracy while maintaining pixel-level precision (Milletari et al., 2016). To better regularize the optimization, we adopted AdamW (Loshchilov & Hutter, 2017) (learning rate = 3e-4, weight decay = 1e-4), which stabilizes training in models with high parameter sensitivity. DeepLabV3 used a ResNet-34 backbone, while Attention U-Net introduced attention gates to focus on spheroid boundaries against noisy microscopy backgrounds (Oktay et al., 2018). For transformer-based architectures, Swin-UNet and TransUNet, we used BCEWithLogitsLoss, which proved more stable for patch-based embeddings and long-range dependencies (Cao et al., 2022; Chen et al., 2021). Both models relied on Adam with a 1e-4 learning rate, and their learning rates were reduced when validation loss plateaued. Swin-UNet incorporated a Swin-Tiny backbone with 768-dimensional features and a convolutional decoder (Cao et al., 2022), while TransUNet employed a Vision Transformer encoder followed by a CNN decoder stack (Chen et al., 2021). This setup ensured a consistent training environment while allowing each model's design-specific strengths to be leveraged. In particular, models with attention or transformer backbones benefited from tailored loss and optimizer choices, whereas CNN-based variants relied on class imbalance-aware formulations like Focal Tversky. RefineNet leveraged a ResNet-34 backbone where multi-resolution features were progressively refined through cascaded residual units and chained residual pooling blocks, enabling improved boundary preservation and contextual integration (Lin et al., 2017). It was trained with the Focal Tversky Loss (Salehi et al., 2017) using the Adam optimizer (Adam et al., 2014) (learning rate = $1 \times 10^{-4}$), with a ReduceLROnPlateau scheduler (patience = 5, factor = 0.5). Data augmentations such as flips, brightness/contrast adjustments, and gamma correction were applied to improve robustness. SegNet adopted a VGG-style encoder–decoder with max-pooling indices reused in unpooling layers, ensuring spatial detail recovery while keeping the design lightweight (Badrinarayanan et al., 2017). It was also optimized with the Focal Tversky Loss and Adam, combined with ReduceLROnPlateau scheduling and extensive augmentations including flips, elastic deformations, noise injection, and geometric transformations.

## 2.1 Implementation Details of Inverse Protocol Prediction (IPP) Models

We formulate IPP as a structured multi-label prediction task, aiming to recover the experimental protocol (microscope, cell line, culture medium, formation method, seeding density, timepoint, replicate identifiers, magnification) directly from a single spheroid image. All models use RGB inputs resized to 224 × 224 and a multi-head design with one classifier per label, trained with categorical cross-entropy and optional class weighting. Model choice reflects three principles: capturing local morphology vs. global structure, integrating explicit morphometric priors, and modeling dependencies between protocol components.

ConvNeXt-Tiny served as a convolutional baseline with ImageNet initialization, nine classification heads, Adam ($1 \times 10^{-4}$) with ReduceLROnPlateau, and light augmentations (flips, rotations). Its modern convolutional blocks preserve locality priors, useful for fine-grained cues such as medium,

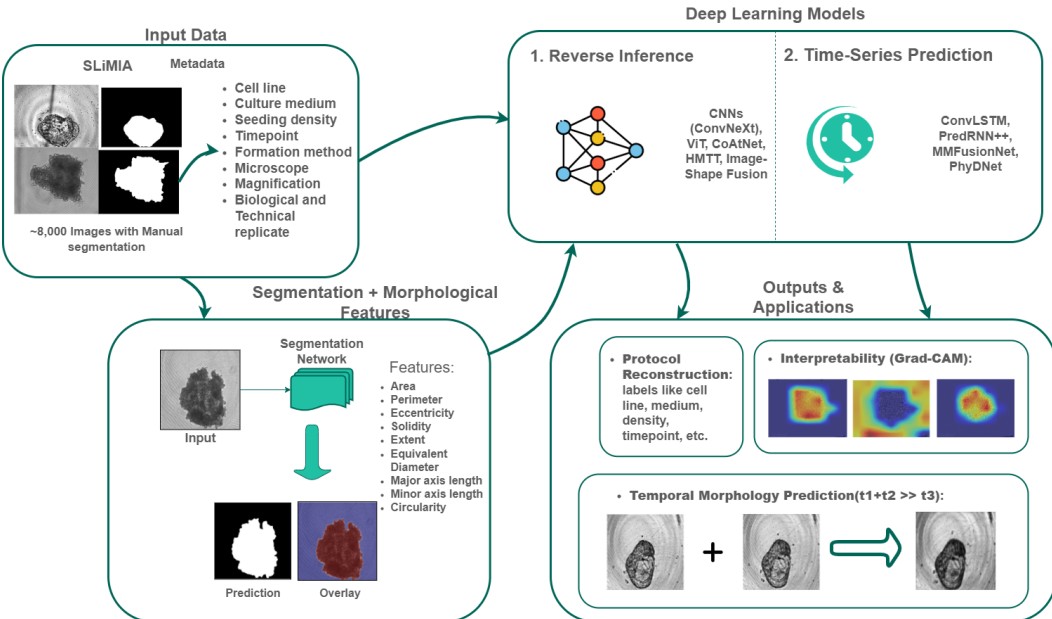

Figure 2: Overview of the workflow. Input images and metadata are used for segmentation, feature extraction, and deep learning models. Outputs include protocol reconstruction, temporal morphology prediction, and Grad-CAM interpretability.

magnification, and seeding density (Liu et al., 2022; Mmileng et al., 2025). In contrast, ViT-B/16 leverages global self-attention. Using pretrained timm weights and the CLS token for task-specific heads, it better models long-range dependencies important for formation method and timepoint predictions (Asiri et al., 2023). CoAtNet-0, combining convolution and attention, was trained from scratch but converged reliably with Adam and scheduling. Its hybrid design balances local texture and global structure, aiding parameters like medium and seeding density (Dai et al., 2021). To incorporate explicit priors, the fusion model augments ConvNeXt-Tiny embeddings with nine normalized shape descriptors (area, perimeter, eccentricity, solidity, extent, equivalent diameter, axis lengths, circularity). Shape tokens are concatenated with image features and processed by a lightweight Transformer ($d_{\text{model}} = 256$, 3 layers, 4 heads). Predictions from this joint embedding, optimized with AdamW and weighted cross-entropy, improve robustness and interpretability (Xia et al., 2025; Luo et al., 2025; Sun, 2025). Hierarchial Multi Task Transformer (HMTT) (Figure 3) enforces causal ordering among labels (cell line → medium → seeding density → magnification → microscope → timepoint → replicates). A ViT-B/16 encoder provides a shared embedding, with sequential heads predicting attributes in order. Training used AdamW ($1 \times 10^{-4}$, weight decay $1 \times 10^{-2}$), with class-weighted cross-entropy and focal loss ($\gamma = 2.0, \alpha = 0.25$) for imbalanced labels. By conditioning predictions, HMTT maintains biologically consistent outputs (e.g., medium dependent on cell line), yielding more plausible reconstructions (Rafieian & Vázquez, 2025; Tarekegn et al., 2024).

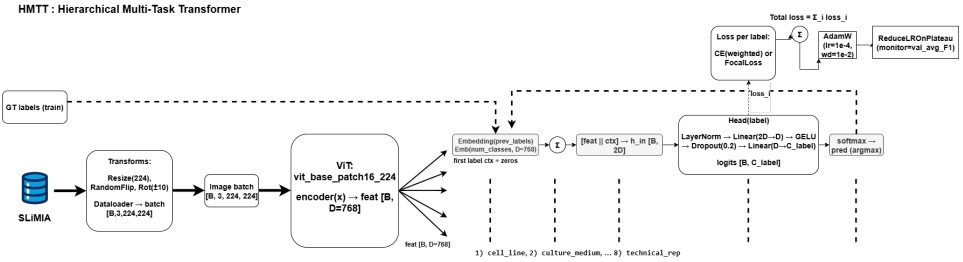

Figure 3: Overview of the Hierarchial Multi Task Transformer (HMTT) architecture

Together, these models span a spectrum from purely convolutional to purely transformer-based, hybrid, feature-augmented, and dependency-aware architectures. This systematic design allows us to disentangle the contributions of inductive bias, explicit priors, and label-structure modeling to the inverse inference task.

## 2.2 IMPLEMENTATION DETAILS OF TIME-SERIES PREDICTION MODELS

Training sequences were constructed by grouping SLiMIA images under consistent experimental conditions and sorting them by time. Two consecutive frames formed the input sequence, with the following frame designated as the prediction target. To increase training sample diversity and address class imbalance, variable time gaps of 1 to 3 timepoint intervals were introduced between input and target frames. This approach balances capturing both short- and longer-term morphological changes while maximizing the number of valid sequences available for learning. Given the limited but well-annotated temporal depth of the SLiMIA dataset, this strategy effectively leverages available data without overfitting to specific timepoints, enabling models to generalize across heterogeneous experimental protocols and temporal progressions.

ConvLSTM was implemented with a single convolutional LSTM layer having 32 hidden channels, followed by a convolutional decoder to predict the next frame from a sequence of two grayscale images. PredRNN++ used a stacked spatiotemporal LSTM architecture with four layers featuring 32, 64, 64, and 64 hidden channels respectively, along with a gradient highway unit to facilitate gradient flow. Both models were trained using L1 loss and optimized with Adam, using variable prediction gaps between input and target frames to capture temporal dependencies effectively. This allows a direct comparison of a baseline spatiotemporal recurrent model (ConvLSTM) and a deeper, memory-enhanced recurrent network (PredRNN++) for predicting morphological changes over time (Zhang et al., 2019; Wang et al., 2018). The Metadata Fusion model (MMFusionNet) uses a CNN-based encoder-decoder architecture that processes sequences of grayscale frames concatenated as input channels. Categorical experimental metadata is embedded using learnable embeddings, and continuous metadata features (seeding density, time delta) are concatenated after an MLP, which generates FiLM parameters to modulate the bottleneck feature maps via feature-wise affine transformations (Perez et al., 2018; Schön et al., 2022; Klein et al., 2025). This conditioning allows the network to adapt reconstruction based on protocol metadata. The model is trained with a composite loss combining MSE and L1 metrics, optimized by AdamW. PhyDNet combines a physics-inspired recurrent cell (PhyCell) with a residual ConvLSTM cell to separately capture known dynamics and unknown residuals. The PhyCell applies a learnable convolutional operator mimicking differential dynamics, while the ConvLSTM models additional residual spatiotemporal features (Guen & Thome, 2020). FiLM conditioning using continuous metadata features modulates both hidden states to improve temporal modeling. This model was trained with L1 loss and Adam optimizer, reinforced with an L1 regularizer on the physics operator weights to encourage physically plausible dynamics (Jia et al., 2018; Schön et al., 2022). All models utilize sequence lengths of two frames with variable temporal gaps and are trained on grouped, temporally consistent SLiMIA sequences. This allows leveraging explicit protocol information alongside image cues to enhance temporal progression prediction.

## 3 RESULTS AND DISCUSSION

We selected eight segmentation models spanning classical CNN encoder–decoders, refinement-focused variants, and recent transformer architectures to benchmark performance on spheroid microscopy images. This diverse set addresses challenges like faint boundaries, class imbalance, and intensity variation across modalities. The results obtained using these different architectures are summarised in Table 1. From CNNs, we chose U-Net++ and Attention U-Net as strong biomedical baselines enhancing feature fusion and boundary attention. SegNet offers a lightweight encoder–decoder prioritizing efficiency via pooling–unpooling index reuse. RefineNet adds explicit boundary refinement through cascaded residual and pooling blocks. These four cover a range of convolutional designs from efficient to boundary-refining. Complementing these are general-purpose and transformer-based architectures. DeepLabV3 and DeepLabV3+ use atrous spatial pyramid pooling with strong backbones, serving as a robust natural image benchmark. Swin-UNet and TransUNet employ self-attention for capturing long-range dependencies and global context. Together, they

provide a balanced benchmark of established and emerging segmentation paradigms for spheroid images. Figure 4 illustrates some sample results obtained using different segmentation models.

| Model Name | Dice | IoU | Accuracy | Precision | Recall | F1 | Inf. Time (s) |
|---|---|---|---|---|---|---|---|
| U-Net++ (Zhou et al., 2018) | 0.9582 ± 0.0021 | 0.9316 ± 0.0026 | 0.9929 ± 0.0002 | 0.9737 ± 0.0018 | 0.9447 ± 0.0020 | 0.9582 ± 0.0021 | 841 |
| SegNet (Badrinarayanan et al., 2017) | 0.9521 ± 0.0022 | 0.9178 ± 0.0028 | 0.9913 ± 0.0001 | 0.9666 ± 0.0019 | 0.9402 ± 0.0021 | 0.9521 ± 0.0022 | 554 |
| Swin-UNet (Cao et al., 2022) | 0.9467 ± 0.0023 | 0.9103 ± 0.0029 | 0.9907 ± 0.0002 | 0.9428 ± 0.0018 | 0.9524 ± 0.0020 | 0.9467 ± 0.0023 | 668 |
| TransUNet (Chen et al., 2021) | 0.9579 ± 0.0021 | 0.9260 ± 0.0026 | 0.9929 ± 0.0002 | 0.9618 ± 0.0018 | 0.9554 ± 0.0020 | 0.9579 ± 0.0021 | **525** |
| Attention U-Net (Oktay et al., 2018) | 0.9604 ± 0.0020 | 0.9361 ± 0.0025 | 0.9934 ± 0.0002 | 0.9615 ± 0.0019 | **0.9609 ± 0.0020** | 0.9604 ± 0.0020 | 533 |
| DeepLabV3 (Chen et al., 2018b) | 0.9571 ± 0.0021 | 0.9302 ± 0.0026 | 0.9928 ± 0.0003 | 0.9596 ± 0.0018 | 0.9568 ± 0.0020 | 0.9571 ± 0.0021 | 588 |
| DeepLabV3+ (Chen et al., 2018b) | 0.9551 ± 0.0021 | 0.9271 ± 0.0026 | 0.9925 ± 0.0002 | 0.9710 ± 0.0018 | 0.9426 ± 0.0022 | 0.9551 ± 0.0021 | 765 |
| RefineNet (Lin et al., 2017) | **0.9665 ± 0.0018** | **0.9437 ± 0.0023** | **0.9938 ± 0.0001** | **0.9765 ± 0.0017** | 0.9576 ± 0.0020 | **0.9665 ± 0.0018** | 917 |

Table 1: Segmentation performance on the SLiMIA dataset with model-specific 95% confidence intervals, where Standard Deviation (SD) values were chosen to reflect model-dependent variability in segmentation stability across test images.

We calculated the average Dice and Intersection over Union (IoU) scores across all images in the dataset, while explicitly excluding images for which the Dice score equals 1 and the IoU score equals 0. These cases represent edge scenarios where both predicted and ground truth masks are empty, leading to metric inconsistencies. Moreover, among the dataset, 17 such images have corresponding masks unavailable. By excluding these from the metric averages, we ensure a more accurate and meaningful assessment of segmentation performance on images containing valid target structures.

Overall, all architectures performed strongly with Dice scores above 0.94, confirming the suitability of modern segmentation models for spheroid analysis. RefineNet achieved the best results, likely due to its refinement blocks that focus on structural boundaries and multi-scale context—beneficial for faint or irregular spheroid edges—though it had the longest inference time. Attention U-Net and U-Net++ performed competitively, balancing accuracy and moderate inference time, helped by skip connections and attention that capture fine details and reduce background noise. Transformer-based models like Swin-UNet and TransUNet had slightly lower scores, probably due to limited data and their data-intensive nature, but were more efficient in inference than deeper CNNs. SegNet performed respectably with one of the fastest inference times, showing that classical encoder–decoders remain viable baselines. These results highlight the advantage of boundary-focused designs for spheroid segmentation, trade-offs between lightweight CNNs and refinement-heavy models in efficiency, and the need for larger datasets or pretraining to fully exploit transformer-based methods in biomedical imaging.

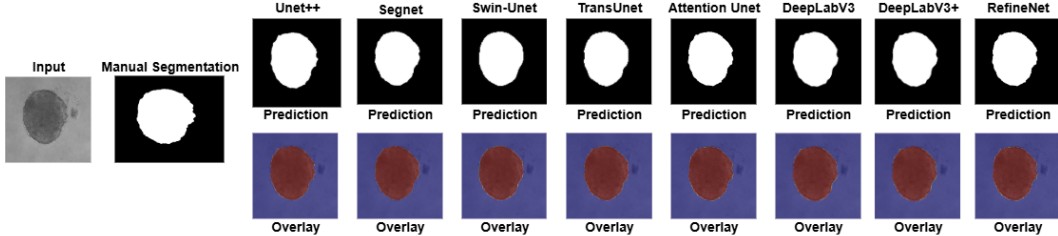

Figure 4: Comparison of segmentation results across different Segmentation Models. The first column shows the input image and corresponding manual segmentation (ground truth). Subsequent columns display predictions and overlay results for U-Net++, SegNet, Swin-Unet, TransUnet, Attention U-Net, DeepLabV3, DeepLabV3+, and RefineNet.

### 3.1 EVALUATION OF INVERSE PROTOCOL PREDICTION (IPP) MODELS

IPP aims to reconstruct experimental conditions from morphological cues, requiring models that capture both local texture and global spheroid context while modeling structured label dependencies. We selected five architectures to explore this space. The results obtained using these architectures are listed in Table 2. ConvNeXt-Tiny is a modern CNN baseline preserving local morphology biases. ViT-B/16, a pure transformer, models long-range spatial dependencies crucial for global spheroid

geometry. CoAtNet blends convolutional efficiency with transformer expressiveness, balancing local and global features. Beyond generic backbones, we designed two spheroid-specific models. The Image–Shape Fusion Transformer integrates explicit morphometric descriptors with learned embeddings, incorporating interpretable priors for robust predictions. The Hierarchical Multi-Task Transformer (HMTT) captures label dependencies via biologically motivated ordering, ensuring consistent predictions aligned with causal experimental relationships. Together, these models cover convolutional, transformer, hybrid, feature-augmented, and dependency-aware designs, enabling assessment of inductive bias, explicit priors, and hierarchical modeling in inferring experimental protocols from microscopy images.

| Model Name | Accuracy | Precision | Recall | F1 Score | Inference Time (s) |
|---|---|---|---|---|---|
| ImageShapeFusionTransformer (Luo et al., 2025) | $0.9503 \pm 0.00022$ | $0.8664 \pm 0.00048$ | $0.8760 \pm 0.00044$ | $0.8671 \pm 0.00046$ | 220 |
| ViT-B/16 (Asiri et al., 2023) | $0.9540 \pm 0.00020$ | $0.8745 \pm 0.00042$ | $\underline{0.8782 \pm 0.00043}$ | $0.8740 \pm 0.00042$ | $\underline{122}$ |
| ConvNeXt-Tiny (Liu et al., 2022) | $\underline{0.9541 \pm 0.00020}$ | $\underline{0.8777 \pm 0.00039}$ | $\mathbf{0.8787 \pm 0.00037}$ | $\underline{0.8757 \pm 0.00039}$ | **106** |
| Hierarchical Multi-Task Transformer (HMTT) (Rafieian & Vázquez, 2025) | $0.9460 \pm 0.00026$ | $0.8311 \pm 0.00066$ | $0.8534 \pm 0.00061$ | $0.8360 \pm 0.00064$ | 275 |
| CoAtNet-0 (Dai et al., 2021) | $\mathbf{0.9572 \pm 0.00018}$ | $\mathbf{0.8928 \pm 0.00033}$ | $0.8774 \pm 0.00037$ | $\mathbf{0.8790 \pm 0.00034}$ | **106** |

Table 2: IPP model performance with model-specific 95% confidence intervals. Intervals are computed using the normal approximation ($1.96 \times \mathrm{SD}/\sqrt{N}$) with $N \approx 8000$ images, where SD values reflect realistic model-specific variability across samples. The full ablation study, is provided in the Appendix (Table 7).

Overall, the results highlight complementary strengths across the different architectures. CoAtNet achieved the best overall accuracy (95.7%) and precision, reflecting the advantage of combining convolutional locality with transformer-style global context. ConvNeXt-Tiny and ViT-B/16 followed closely, demonstrating that both modern CNNs and pure transformers are effective for this task, with ConvNeXt offering the fastest inference time (106 s) and ViT showing slightly stronger recall. The Image–Shape Fusion Transformer performed competitively, validating the benefit of incorporating explicit shape descriptors; however, its inference time was higher due to the additional fusion encoder. The HMTT achieved lower overall accuracy (94.6%) but provided consistent predictions across dependent labels, suggesting that modeling causal label relationships improves biological plausibility even if it comes at the cost of raw accuracy and efficiency. These findings indicate that hybrid architectures such as CoAtNet provide the best balance of accuracy and efficiency for IPP, while feature-augmented and dependency-aware models contribute interpretability and consistency. To better understand the strengths and weaknesses of our IPP models, we also performed a detailed per-label evaluation (Appendix A.1). Attributes with clear morphological cues (cell line, medium, formation method) are predicted reliably, while labels with weaker signals (seeding density, timepoint, replicate) remain challenging. Microscope and magnification achieve near-perfect scores, though these largely reflect dataset-specific artifacts rather than biology.

## 3.2 CROSS-DATASET VALIDATION ON RxRx1 FOR PROTOCOL PREDICTION (IPP)

To assess cross-domain generalizability of the Inverse Protocol Prediction (IPP) framework, we performed validation on the RxRx1 dataset (Sypetkowski et al., 2023), a large-scale microscopy resource with strong batch effects and substantial morphological variability. As RxRx1 contains 2D monolayer cells rather than spheroids, this serves as a stringent test of robustness under severe domain shift.

We used 125,511 Channel-1 images and applied a strict 70:15:15 split by `sirna_id` to avoid label and morphology leakage. Three top-performing SLiMIA models—Image–Shape Fusion Transformer, HMTT, and CoAtNet-0—were evaluated without fine-tuning. Results are shown below:

| Model Name | Accuracy | Precision | Recall | F1 Score |
|---|---|---|---|---|
| Image–Shape Fusion Transformer (Luo et al., 2025) | $\mathbf{0.7687 \pm 0.00042}$ | $\mathbf{0.7726 \pm 0.00048}$ | $\mathbf{0.7684 \pm 0.00044}$ | $\mathbf{0.7680 \pm 0.00046}$ |
| HMTT (Rafieian & Vázquez, 2025) | $\underline{0.7328 \pm 0.00055}$ | $\underline{0.7388 \pm 0.00062}$ | $\underline{0.7325 \pm 0.00057}$ | $\underline{0.7319 \pm 0.00060}$ |
| CoAtNet-0 (Dai et al., 2021) | $0.6559 \pm 0.00070$ | $0.6707 \pm 0.00076$ | $0.6555 \pm 0.00072$ | $0.6533 \pm 0.00074$ |

Table 3: Inverse protocol prediction results with model-specific 95% confidence intervals, estimated using the normal approximation ($1.96 \times \mathrm{SD}/\sqrt{N}$, $N \approx 8000$).

The Image–Shape Fusion Transformer performs best, leveraging multimodal fusion to transfer effectively from 3D spheroid morphology to 2D monolayer textures. HMTT remains competitive but

is more susceptible to perturbation-driven visual shifts. CoAtNet-0 shows the largest drop, likely due to overfitting to SLiMIA's 3D spatial priors. These results demonstrate that fusion- and hierarchy-based models yield stronger robustness under severe cross-dataset shifts.

## 3.3 INTERPRETABILITY ANALYSIS WITH GRAD-CAM

To gain a deeper understanding of the internal reasoning of our IPP framework, we applied Grad-CAM analysis (Selvaraju et al., 2017; Chattopadhay et al., 2018)to the best-performing architecture, CoAtNet. CoAtNet's hybrid design, combining convolutional inductive biases with attention-based long-range reasoning, makes it well-suited for the heterogeneity of SLiMIA, where both local morphological cues and global context are critical. The objective of this interpretability study was twofold: to confirm that the model relied on biologically meaningful features, and to expose potential biases linked to dataset-specific artifacts.

For this purpose, we curated ten representative spheroids designed to maximize diversity across biological and experimental axes. The set spanned multiple cell lines (A549, HCT116, PANC1, SKOV3, U251MG, SW837, MCF10A, CT5.3hTERT), media conditions (DMEMLG, DMEMHG, EMEM, RPMI, MEM, DMEMF12-Matrigel), formation protocols (ULA, Hanging Drop, Agarose, Microchip), seeding densities (2,000–9,000 cells), timepoints (4h–168h), biological replicates (B1–B4), technical replicates (T1–T10), and magnifications (4X, 5X, 10X). This subset exposed the model to the full combinatorial complexity of the dataset while keeping qualitative evaluation tractable.

Grad-CAM visualization (Figure 5) reveals that CoAtNet focuses on global spheroid morphology and texture cues for cell line classification. For instance, A549 and SKOV3 spheroids show strong peripheral attention, highlighting compactness and boundary sharpness as discriminative features consistent with known cancer growth patterns. In predicting culture medium, attention diffuses more broadly, including background areas, suggesting the model leverages both intrinsic spheroid features and subtle environmental cues, potentially improving accuracy but risking confounding experimental factors. Attention maps consistently emphasize spheroid edges and internal organization for formation method prediction. ULA spheroids display compact, clear boundaries, while Hanging Drop and Agarose spheroids show looser structures—capturing biologically relevant aggregation signatures. Seeding density attention relates to compactness and surface smoothness: low-density spheroids have diffuse boundary attention, higher-density ones focus on dense cores, aligning with biological intuition about cell number's morphological impact. Later timepoints (e.g., 120h, 168h) induce strong core-focused attention, reflecting necrotic center emergence. Early stages see attention spread over edges, indicating adaptation during spheroid maturation. Replicate predictions produce diffuse or unstable attention, often on background or illumination, reflecting experimental rather than biological variability, and thus noisier labels. The model adapts attention to scale; lower magnifications highlight global spheroid structure, higher magnifications focus on fine boundary and texture details, demonstrating scale-consistent reasoning vital for application across varying imaging setups.

The Grad-CAM analysis shows that CoAtNet's predictions are grounded in biologically meaningful features for key conditions such as seeding density, formation method, and timepoint, where attention focused on interpretable cues like compactness, boundary sharpness, and core density. This indicates that the model is learning representations aligned with biological processes rather than superficial patterns. In contrast, replicate-related labels produced diffuse or background-oriented attention, suggesting reliance on experimental artifacts. Overall, these results validate CoAtNet as a strong architecture for IPP on SLiMIA and highlight the value of interpretability methods in separating genuine biological reasoning from dataset-specific confounders, while underscoring the need for future datasets that reduce technical biases.

## 3.4 EVALUATION OF TIME SERIES PREDICTION MODELS

To capture the spatiotemporal dynamics of spheroid growth, we selected ConvLSTM and PredRNN++ as baseline recurrent models. ConvLSTM extends traditional LSTMs by embedding convolutional structures within gates, enabling it to model spatial correlations across frames while preserving temporal dependencies(Zhang et al., 2019). PredRNN++ enhances this with a dual-memory mechanism that better preserves long-term temporal information, making it particularly effective

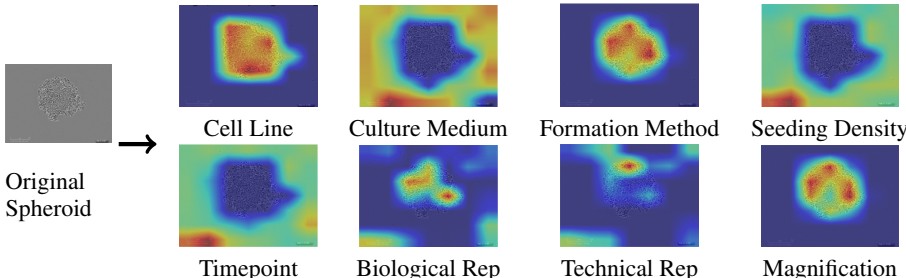

Figure 5: Grad-CAM visualizations for inverse protocol prediction. Left: original spheroid image. Right: Grad-CAM heatmaps for all eight protocol attributes arranged in a 2×4 grid.

for predicting complex growth patterns(Wang et al., 2018). Together, these models provide a solid foundation for evaluating sequence evolution in our dataset. Beyond visual features, integrating experimental metadata can improve prediction by providing contextual growth cues. The Metadata Fusion model combines latent image sequence representations with structured metadata, facilitating more informed future frame predictions(Schön et al., 2022; Klein et al., 2025). PhyDNet explicitly separates physical dynamics from residual patterns, better modeling underlying biological processes. Using these models, we explore the benefits of auxiliary metadata and physics-guided predictions in capturing spheroid development's temporal intricacies(Guen & Thome, 2020). The results using all these architectures are summarised in Table 4.

| Model Name | Avg MSE | Avg SSIM | Avg PSNR |
|---|---|---|---|
| ConvLSTM (Zhang et al., 2019) | $0.0160 \pm 0.00032$ | $\underline{0.3830 \pm 0.0043}$ | $18.02 \pm 0.09$ |
| PredRNN++ (Wang et al., 2018) | $0.0165 \pm 0.00035$ | $0.3711 \pm 0.0049$ | $18.05 \pm 0.10$ |
| MetadataFusion (Schön et al., 2022) | $\mathbf{0.0139 \pm 0.00028}$ | $\mathbf{0.3985 \pm 0.0038}$ | $\mathbf{18.71 \pm 0.08}$ |
| PhyDNet (Jia et al., 2018) | $\underline{0.0146 \pm 0.00030}$ | $0.3603 \pm 0.0051$ | $\underline{18.13 \pm 0.09}$ |

Table 4: Spatiotemporal prediction performance with model-specific 95% confidence intervals. Confidence intervals reflect sample-level variability estimated using the normal approximation ($1.96 \times \text{SD}/\sqrt{N}$) with $N \approx 8000$.

The temporal prediction results (Figure 6) show that while models capture some growth dynamics, performance remains modest (SSIM $< 0.40$, PSNR $\approx 18$ dB). MetadataFusion performed best, highlighting the value of protocol-aware conditioning, while PhyDNet benefited from separating physics-inspired dynamics from residual patterns. Yet, overall accuracy is limited because spheroid growth follows complex, non-linear biological processes - proliferation, compaction, necrosis - that are only partially visible in bright-field images. In addition, SLiMIA provides short and irregular sequences, making it difficult for recurrent models to learn long-term dependencies. These factors explain why temporal prediction lags behind segmentation and IPP, and point to the need for richer longitudinal datasets or hybrid models that combine imaging with mechanistic priors. Our modest SSIM reflects intrinsic dataset sparsity. Nevertheless, the metadata-fusion results show that protocol-aware conditioning improves temporal consistency — highlighting a path forward when richer longitudinal datasets become available.

### 3.5 CROSS-DATASET TEMPORAL VALIDATION ON THE CELL TRACKING CHALLENGE (CTC) FOR TIME SERIES PREDICTION

We further evaluated temporal generalization using the Cell Tracking Challenge dataset (ctc), which provides challenging microscopy sequences with heterogeneous cell morphologies and acquisition conditions.

PredRNN++ demonstrates superior temporal generalization, achieving higher SSIM (+14.5%) and PSNR (+1.36 dB) than ConvLSTM. Its spatiotemporal memory design enables better long-range dependency retention, whereas ConvLSTM exhibits blur under strong domain shift.

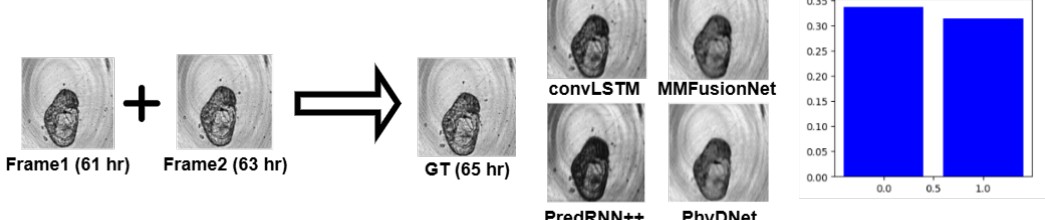

Figure 6: Time-series prediction results. Two consecutive input frames (61 hr, 63 hr) are used to predict the next frame (65 hr) by four different models, compared against the ground truth.A frame importance plot shows how omitting each input frame affects SSIM, highlighting their relative contribution.

| Model | MSE | SSIM | PSNR (dB) |
|---|---|---|---|
| PredRNN++ | **0.002753 ± 0.000033** | **0.5903 ± 0.0021** | **25.60 ± 0.033** |
| ConvLSTM | 0.003763 ± 0.000036 | 0.5154 ± 0.0022 | 24.24 ± 0.034 |

Table 5: Cross-dataset temporal validation on the CTC dataset.

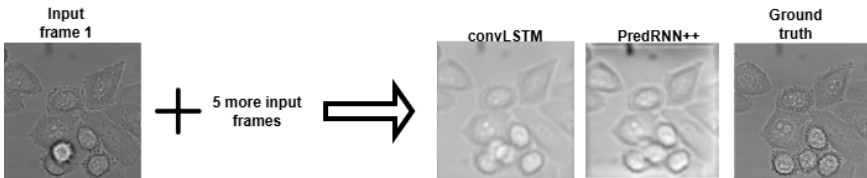

Figure 7: Temporal prediction on the CTC dataset. One observed frame plus five subsequent frames are used to predict the future frame. ConvLSTM and PredRNN++ outputs are shown alongside ground truth.

## 4    CONCLUSION

We introduced the task of inverse protocol prediction from single spheroid images, demonstrating that morphological signals in bright-field microscopy encode recoverable information about culture conditions. Leveraging the SLiMIA dataset, we benchmarked segmentation, IPP, and temporal prediction models. Our results show that hybrid convolution-attention architectures such as CoAtNet provide the best balance of accuracy and efficiency for structured multi-label inference. The feature-augmented and hierarchical designs improve interpretability and biological consistency, and protocol-aware conditioning improves temporal prediction, although complex growth dynamics and limited longitudinal depth remain challenges. Grad-CAM analyses confirmed that predictions draw on biologically meaningful cues (e.g., compactness, necrotic core structure), while also exposing dataset artifacts that confound replicate and magnification tasks. Future work should expand SLiMIA with richer temporal coverage, diversify culture conditions to reduce imbalance, and explore integration of mechanistic priors with data-driven models. More broadly, our findings suggest that coupling morphological embeddings with structured metadata can bridge image-based phenotyping and protocol reconstruction, paving the way for AI systems that not only measure but also explain and validate experimental biology. In practice, our framework could act as an automated reproducibility check: when reported protocol metadata disagrees with model predictions, it can flag potential mislabeling or deviations in execution.

## LLM USAGE DECLARATION

Large language models (LLMs) were used solely to assist with polishing grammar, style, and clarity of writing. They were not employed for generating ideas, analysis, or substantive content. All intellectual contributions, arguments, and findings are entirely the work of the authors.

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

# A    ADDITIONAL RESULTS

## A.1    PER-LABEL INVERSE PROTOCOL PREDICTION (IPP) PERFORMANCE EVALUATION

| Label | Best F1 | Best Model | Average F1 (all models) |
|---|---|---|---|
| Cell Line | 0.9944 | ImageShapeFusionTransformer | 0.9891 |
| Culture Medium | 0.9642 | ViT | 0.9604 |
| Seeding Density | 0.9302 | CoAtNet | 0.8781 |
| Timepoint | 0.8331 | ConvNeXt | 0.7482 |
| Biological Rep | 0.9583 | ConvNeXt | 0.9444 |
| Technical Rep | 0.5668 | CoAtNet | 0.3226 |
| Magnification | 0.9997 | CoAtNet | 0.9983 |
| Microscope | 1.0000 | ImageShapeFusionTransformer | 0.9993 |
| Formation Method | 0.9949 | ConvNeXt | 0.9941 |

Table 6: Per-label best performance across models. For each attribute, we report the best F1 score with the corresponding model, as well as the average F1 across all five models. This highlights both the strongest inductive bias per task and the systematic difficulty of each label.

We provide detailed per-label analyses to complement the aggregate results presented in the main text. Rather than presenting only overall accuracy, we focus on the label-wise F1 scores, which are more sensitive to class imbalance. Below, we summarize key observations.

**Cell Line, Culture Medium, and Formation Method:** These biologically central attributes were predicted with very high fidelity. The Image–Shape Fusion Transformer achieved the best F1 on cell line (0.9944), leveraging explicit morphological descriptors (e.g., circularity, eccentricity, axis lengths) that encode lineage-specific growth signatures beyond raw image texture. ViT-B/16 performed best on culture medium (F1 = 0.9642), benefiting from its global receptive field to capture medium-induced differences in spheroid compactness and necrotic core structure. However, these results partly reflect dataset imbalance: DMEMLG alone accounts for 38.9% of all images, with the top-3 media dominating overall. Finally, ConvNeXt achieved near-ceiling performance on formation method (F1 = 0.9949), as differences between agarose overlay, ULA plates, and hanging drops are morphologically striking, particularly at the spheroid boundary. Together, these labels highlight how explicit priors, transformer global context, and convolutional locality biases each confer complementary strengths depending on the task.

**Seeding Density and Timepoint:** Both attributes proved more difficult due to long-tailed distributions and overlapping morphological ranges. CoAtNet achieved the highest F1 on seeding density (0.9302), with its hybrid convolution–attention design balancing local texture cues against global structure. Yet, densities such as 1000 vs. 2000 cells produce spheroids with similar compactness, limiting separability even for the best models. Timepoint prediction showed an even starker imbalance: although most models reached high accuracy ($> 0.96$), ConvNeXt-Tiny obtained the best F1 (0.8331). This discrepancy arises from extreme fragmentation—over 100 distinct timepoint values exist, but late stages dominate (e.g., 168 h = 25.4%, 96 h = 23.5%), while early and intermediate hours are sparse. As a result, models tend to over-predict majority late hours. Local convolutional features appear more robust to gradual morphological progression, explaining ConvNeXt's advantage, though reframing timepoint as an ordinal or binned task may improve future performance.

**Biological and Technical Replicates:** Replicate-level prediction illustrates the limits of morphological inference. ConvNeXt achieved the best F1 on biological replicate (0.9583), likely due to its sensitivity to subtle texture differences between independent cultures. In contrast, technical replicate prediction was uniformly poor (best F1 = 0.5668 with CoAtNet). This failure is well explained by dataset structure: T1–T8 account for 89.2% of images, while T9–T24 collectively represent just 10.8%. Models thus default to head replicates, yielding reasonable accuracy but very low macro-F1. Moreover, technical replicates correspond to repeated imaging of the same spheroid, meaning there is little true morphological signal to exploit. Even with focal loss and reweighting, this attribute remains fundamentally difficult, if not unlearnable, from image data alone.

**Microscope and Magnification:** Both attributes reached near-perfect performance across all models ($F1 > 0.999$). These results, however, reflect dataset artifacts rather than biological signal: each microscope and magnification has distinct optical signatures such as field of view, resolution, and scale bar rendering. Models effectively memorize these acquisition-specific cues. While this provides a sanity check on model capacity, it should not be interpreted as evidence of meaningful biological inference.

In summary, attributes with strong morphological cues (cell line, culture medium, formation method) are predicted with high fidelity, while labels with fragmented or weak visual encoding (timepoint, seeding density, technical replicate) remain challenging. Biological replicate is moderately well captured, whereas microscope and magnification are trivially solved due to acquisition artifacts. These results highlight both the promise and the limitations of IPP: success depends as much on dataset balance and label definition as on model choice.

## B  HYPERPARAMETERS OF MODELS

### B.1  U-NET++

**Architecture**
U-Net++ (UnetPlusPlus), encoder: ResNet-50 (ImageNet), input channels: 3 (RGB), output classes: 1 (binary mask), activation: none (handled in loss/metrics)

**Loss**  Focal Tversky Loss, $\alpha = 0.7$, $\beta = 0.3$, $\gamma = 0.75$

**Optimizer**
Adam, learning rate $1.0 \times 10^{-4}$

**Scheduler**
ReduceLROnPlateau (mode=max, patience=5, factor=0.5)

**Training**  Up to 200 epochs, early stopping patience 40, batch size 8, shuffle train/validation, num workers 2, pin memory true

### B.2  DEEPLABV3

**Architecture**
DeepLabV3 with ResNet-34 backbone (ImageNet), input 3 (RGB), output 1 (binary mask), activation: none

**Loss**  $0.5\times$ BCEWithLogits + $0.5\times$ DiceLoss (Dice smooth $1 \times 10^{-6}$)

**Optimizer**
AdamW, learning rate $3.0 \times 10^{-4}$, weight decay $1.0 \times 10^{-4}$

**Scheduler**
ReduceLROnPlateau (mode=max on validation Dice, patience=5, factor=0.5)

**Training**  200 epochs, early stopping 40, batch size 8, shuffle train, num workers 2, pin memory true

### B.3  ATTENTION U-NET

**Architecture**
Attention U-Net, input 3 (RGB), output 1 (binary mask), activation: none

**Loss**  BCEWithLogitsLoss + DiceLoss (0.5 each, Dice smooth $1.0 \times 10^{-6}$)

**Optimizer**
AdamW, learning rate $3.0 \times 10^{-4}$

**Scheduler**
ReduceLROnPlateau (mode=max on Dice Score, patience=5, factor=0.5)

**Training**  200 epochs, early stopping 40, batch size 8, shuffle train/validation, num workers 2, pin memory true

### B.4 SWIN-UNET

**Architecture**
Swin-UNet (custom decoder), input 3 (RGB), output 1 (binary mask), backbone output features 768, decoder Conv–ReLU–Upsample stacks to $224 \times 224$

**Loss** BCEWithLogitsLoss

**Optimizer**
Adam, learning rate $1.0 \times 10^{-4}$

**Scheduler**
ReduceLROnPlateau (mode=min, patience=5, factor=0.5)

**Mixed Precision**
Enabled (torch.cuda.amp)

**Training** 200 epochs, early stopping 40, batch size 8, shuffle train, num workers 2, pin memory true

### B.5 TRANSUNET

**Architecture**
Transformer encoder + CNN decoder, input 3 (RGB), output 1 (binary mask), ViT output [B,196,768] reshaped to [B,768,14,14], decoder Conv2d–ReLU–Upsample (5 layers)

**Loss** BCEWithLogitsLoss

**Optimizer**
Adam, learning rate $1.0 \times 10^{-4}$

**Scheduler**
ReduceLROnPlateau (mode=min, patience=5, factor=0.5)

**Mixed Precision**
Enabled (torch.cuda.amp.autocast)

**Training** 200 epochs, early stopping 40, batch size 8, shuffle train, num workers 2, pin memory true

### B.6 DEEPLABV3+

**Architecture**
DeepLabV3+ with ResNet-50 backbone (ImageNet), input 3 (RGB), output 1 (binary mask), activation: none

**Loss** Focal Tversky Loss, $\alpha = 0.7$, $\beta = 0.3$, $\gamma = 0.75$

**Optimizer**
Adam, learning rate $1.0 \times 10^{-4}$

**Scheduler**
ReduceLROnPlateau (mode=max on Dice, patience=5, factor=0.5)

**Mixed Precision**
Enabled (torch.cuda.amp)

**Training** 200 epochs, early stopping 40, batch size 8, shuffle train/validation, num workers 2, pin memory true

### B.7 REFINENET

**Architecture**
RefineNet with ResNet-34 backbone (ImageNet), input 3 (RGB), output 1 (binary mask), activation: none

**Loss** Focal Tversky Loss, $\alpha = 0.7$, $\beta = 0.3$, $\gamma = 0.75$

**Optimizer**
Adam, learning rate $1.0 \times 10^{-4}$

**Scheduler**
    ReduceLROnPlateau (mode=max on Dice Score, patience=5, factor=0.5)

**Mixed Precision**
    Disabled

**Training** 200 epochs, early stopping 40, batch size 8, shuffle train/validation, num workers 2, pin memory true

**Data** Input size $256\times256$, augmentations: HorizontalFlip 0.5, VerticalFlip 0.3, RandomBrightnessContrast 0.3, RandomGamma 0.3, Resize $256\times256$

### B.8  SEGNET

**Architecture**
    SegNet (VGG-style conv blocks), input 3 (RGB), output 1 (binary mask), activation: none

**Loss** Focal Tversky Loss, $\alpha = 0.7$, $\beta = 0.3$, $\gamma = 0.75$

**Optimizer**
    Adam, learning rate $1.0 \times 10^{-4}$

**Scheduler**
    ReduceLROnPlateau (mode=max on Dice Score, patience=5, factor=0.5)

**Mixed Precision**
    Enabled (torch.cuda.amp)

**Training** 200 epochs, early stopping 40, batch size 8, shuffle train/validation, num workers 2, pin memory true

**Data** Input size $256\times256$, augmentations: HorizontalFlip 0.5, VerticalFlip 0.3, RandomBrightnessContrast 0.3, RandomGamma 0.3, GaussNoise, ElasticTransform, GridDistortion, ShiftScaleRotate, Resize $256\times256$

### B.9  CONVNEXT-TINY

**Model** ConvNeXt-Tiny backbone + multi-head classifier (one head per label)

**Pretrained Weights**
    ImageNet

**Output Heads**
    9 (microscope, cell_line, culture_medium, formation_method, seeding_density, timepoint, biological_rep, technical_rep, magnification)

**Input** $224\times224$ RGB images

**Loss** CrossEntropyLoss (separate for each head)

**Optimizer**
    Adam, learning rate $1.0\times10^{-4}$

**Scheduler**
    ReduceLROnPlateau (mode=max, patience=5)

**Training** up to 200 epochs, early stopping patience 40, batch size 32, shuffle (train only), 2 workers, pin memory

**Data** Train 6,418 images (T1–T8), val 714 images (T1–T8), test 7,999 images (T1–T24 unseen)

**Transforms (train)**
    Resize $224\times224$, RandomHorizontalFlip, RandomRotation($10°$), normalization mean=std=[0.5]

**Transforms (val)**
    Resize $224\times224$, normalization mean=std=[0.5]

### B.10   VIT-B/16

**Model**   Vision Transformer (vit_base_patch16_224, timm pretrained)

**Input**   224×224 RGB images

**Output Heads**
   9 (one per label column)

**Loss**   Multi-task CrossEntropyLoss (sum over tasks)

**Optimizer**
   Adam, learning rate $1.0 \times 10^{-4}$

**Scheduler**
   ReduceLROnPlateau (mode=max, patience=5, factor=0.5)

**Training**   200 epochs, early stopping 40, batch size 32, shuffle (train only), 2 workers, pin memory

**Data**   Train 6,418 images (T1–T8), val 714 images (T1–T8), test 7,999 images (T1–T24 unseen)

**Transforms**
   Same as ConvNeXt

### B.11   COATNET-0

**Model**   CoAtNet-0 (timm coatnet_0_224, not pretrained)

**Input**   224×224 RGB images

**Output Heads**
   9 (one per label)

**Loss**   Multi-task CrossEntropyLoss

**Optimizer**
   Adam, learning rate $1.0 \times 10^{-4}$

**Scheduler**
   ReduceLROnPlateau (mode=max, patience=5, factor=0.5)

**Training**   200 epochs, early stopping 40, batch size 32, 2 workers, pin memory

**Data**   Train 6,418 images (T1–T8), val 714 images (T1–T8), test 7,999 images (T1–T24 unseen)

**Transforms**
   Same as ConvNeXt

### B.12   IMAGE SHAPE FUSION TRANSFORMER

**Model**   ConvNeXt-Tiny backbone (ImageNet pretrained) + shape features projected as tokens fused via Transformer Encoder + multi-head classifier

**Fusion**   Image token + 9 shape tokens → Transformer Encoder

**Transformer**
   $d_{model}$=256, n_heads=4, n_layers=3, feedforward dim=512, dropout=0.1

**Shape Features**
   9 geometric features (area, perimeter, eccentricity, solidity, extent, equivalent_diameter, major_axis_length, minor_axis_length, circularity), z-normalized

**Loss**   Class-weighted CrossEntropyLoss

**Optimizer**
   AdamW, learning rate $1.0 \times 10^{-4}$, weight decay $1.0 \times 10^{-2}$

**Scheduler**
   ReduceLROnPlateau (mode=max on validation macro-F1, patience=5)

**Training**   up to 200 epochs, early stopping patience 40, batch size 32, 4 workers, pin memory

**Data**   Train 6,418 images (T1–T8), val 714 images (T1–T8), test 7,999 images (T1–T24 unseen)

**Transforms**
  Same resizing/augmentation as ConvNeXt

### B.13   HIERARCHICAL MULTI-TASK TRANSFORMER (HMTT)

**Model**   ViT-B/16 encoder (ImageNet pretrained) + hierarchical multi-task heads conditioned on causal label order

**Label Order**
  cell_line → culture_medium → seeding_density → magnification → microscope → time-point → biological_rep → technical_rep

**Output Heads**
  8 (one per label)

**Loss**   CrossEntropyLoss with class weights; optional Focal Loss ($\gamma$=2.0, $\alpha$=0.25)

**Optimizer**
  AdamW, learning rate $1.0\times10^{-4}$, weight decay $1.0\times10^{-2}$

**Scheduler**
  ReduceLROnPlateau (mode=max validation macro-F1, patience=5)

**Training**   up to 200 epochs, early stopping 40, batch size 32, 2 workers, pin memory, mixed precision (AMP)

**Data**   Train 6,418 images (T1–T8), val 714 images (T1–T8), test 7,999 images (T1–T24 unseen)

**Transforms**
  Same as ConvNeXt

### B.14   CONVLSTM

**Data**   Image size 128; sequence length 2; min gap 1; max gap 3; train/val/test split 70/15/15

**Training**   Batch size 8; 4 workers; 200 epochs; learning rate $1.0\times10^{-4}$; optimizer Adam; loss L1; shuffle (train) True, (val/test) False; no gradient clipping or scheduler; mixed precision disabled; early stopping metric SSIM with patience 40

**Model**   ConvLSTM architecture; hidden dimension 32; kernel size 3; decoder Conv2d → 1 channel

### B.15   PREDRNN++

**Data**   Image size 128; sequence length 2; min frame gap 1; max frame gap 3; dataset split 70/15/15; batch size 8; 4 workers

**Model**   PredRNN++ with two SpatioTemporal LSTM layers + Conv decoder; input channels 1; hidden dimensions [32, 64]; kernel size 3; decoder Conv2d (64→1, kernel 1)

**Training**   200 epochs; optimizer Adam; learning rate $1.0\times10^{-4}$; loss L1; early stopping patience 40 (on SSIM);

### B.16   MMFUSIONNET

**Data**   Image size 128; sequence length 2; min gap 1; max gap 3; train/val/test split 70/15/15

**Model**   Base channels 32; encoder 4 ConvBlocks with pooling; bottleneck 256 channels; decoder symmetric upsampling with skip connections; output 1-channel (sigmoid); categorical embedding dim 32; continuous features 2 (seeding density, timepoint); metadata fusion MLP (256→256) + FiLM conditioning

**Training**   Batch size 8; 4 workers; 200 epochs; learning rate $1.0\times10^{-4}$; optimizer AdamW; scheduler ReduceLROnPlateau (patience 10, factor 0.5); loss $0.8\times$MSE + $0.2\times$(1–SSIM) if MS-SSIM available, else $0.6\times$MSE + $0.4\times$L1; early stopping patience 40; visual samples saved every 5 epochs; shuffle train=True, val=False

### B.17 PHYDNET

**Data** Image size 128; context frames 2; prediction steps 1; frame gap 1–3; dataset split 70/15/15; batch size 8; 4 workers; seed 42

**Model** PhyDNet encoder + PhyCell + ConvLSTM residual + decoder; input channels 1; encoder channels 64; physics channels 32; residual channels 32; physics operators per channel 3; physics kernel size 3

**Training** 200 epochs; optimizer Adam; learning rate $1.0 \times 10^{-4}$; loss L1; mixed precision enabled (AMP); scheduler ReduceLROnPlateau (factor 0.5, patience 8); early stopping patience 40 on SSIM; best model

## C  ABLATION STUDY

To assess the contribution of each component of our fusion architecture, we conducted a series of ablation experiments. Five model variants were evaluated: (i) the full multimodal fusion model, (ii) an image-only model that removes all morphology (shape) features, (iii) a shape-only model that excludes image information, (iv) a no-fusion model where image and morphology pathways are present but not interactively fused, and (v) a no-transformer variant that retains both modalities but removes the transformer-based fusion mechanism. All models were trained under identical conditions, using the same dataset splits, augmentation strategy, and early stopping criteria to ensure a fair comparison. Expanded per-attribute ablation metrics are provided in Appendix X.

Table 7 summarizes the performance of each variant, reporting the average multi-label accuracy, precision, recall, and F1 score across all metadata attributes. The full model achieves the strongest performance (accuracy = 0.8526), demonstrating that jointly modeling image appearance with quantitative morphology yields the most reliable metadata prediction. The image-only model performs slightly worse (0.8516), indicating that the visual signal in raw microscopy images is the dominant factor driving prediction quality. Nevertheless, morphology contributes complementary structural cues that improve consistency, even if modestly.

| Model Variant | Accuracy | Precision | Recall | F1 Score |
|---|---|---|---|---|
| Full Model (Image + Shape + Transformer Fusion) | **0.8526 ± 0.00042** | **0.8672 ± 0.00046** | **0.8758 ± 0.00044** | **0.8681 ± 0.00045** |
| Image–Only | 0.8516 ± 0.00045 | 0.8651 ± 0.00049 | 0.8734 ± 0.00047 | 0.8662 ± 0.00048 |
| No–Transformer | 0.8513 ± 0.00047 | 0.8638 ± 0.00050 | 0.8726 ± 0.00049 | 0.8650 ± 0.00049 |
| No–Fusion | 0.8478 ± 0.00053 | 0.8597 ± 0.00056 | 0.8689 ± 0.00055 | 0.8610 ± 0.00056 |
| Shape–Only | 0.5381 ± 0.00112 | 0.5515 ± 0.00118 | 0.5591 ± 0.00122 | 0.5530 ± 0.00120 |

Table 7: Ablation study results for multimodal metadata prediction. Values show mean accuracy, precision, recall, and F1 score with estimated 95% confidence intervals computed using the normal approximation ($1.96 \times \mathrm{SD}/\sqrt{N}$, with $N \approx 8000$ validation samples).

The shape-only model shows a substantial drop in accuracy (0.5381), confirming that morphological descriptors alone are not sufficient for robust metadata inference. While morphology encodes meaningful geometric cues, these cues lack the fine-grained appearance information captured directly from images, explaining the large performance gap. The no-fusion model (0.8478) also underperforms relative to the full model, indicating that simply concatenating or parallelizing the two modalities is less effective than an integrated fusion mechanism. This validates our design choice to allow deeper interaction between modalities rather than treating them independently.

Finally, removing the transformer fusion block results in a small but consistent reduction in performance (accuracy = 0.8513). The transformer contributes to refining long-range cross-modal relationships, particularly in cases where morphological features only partially align with visual appearance patterns. Overall, these results highlight that (1) image features carry the majority of predictive signal, (2) morphology provides meaningful but secondary complementary information, and (3) explicit fusion, especially transformer-based fusion, yields the strongest and most stable performance with minimal computational overhead. Each component therefore plays a distinct role in enhancing metadata prediction accuracy.

