# OpenReview forum: "One Snapshot, Many Clues: Inverse Protocol Prediction from Single-View Spheroid Images"
_ICLR.cc/2026/Conference — Submitted to ICLR 2026_

### Official Review · Reviewer_9dmv · 2025-11-03

**Soundness:** 3
**Presentation:** 1
**Contribution:** 2
**Rating:** 4
**Confidence:** 4

**Summary:**

This paper introduces the task of inverse protocol prediction which is the problem of inferring experimental metadata such as cell line, edium, seeding density, timepoint, microscopy, etc... from a single spheroid image. The authors use the publicly available SLiMIA dataset to benchmark different model types for this task. They further propose two custom designs.

**Strengths:**

* Interesting new problem with potential uses in science and industry
* Clearly described experimental setup

**Weaknesses:**

* reverse inference is a problematic term as it is an established term in neuroscience and also has misleading implications in an ML context
* The task of inverse protocol prediction is essentially multi-label metadata classification. The framing as a new “inverse modeling” paradigm is overstated
* The model likely captures acquisition-specific signatures more than biologically meaningful morphology - while this can also be interesting it points to less new biological understanding than implied

**Questions:**

Consider renaming the reverse inference task - maybe Inverse Protocol Prediction which I think you mentioned in the paper or Protocol Inference

---

> ### Author Response · Authors · 2025-11-22
>
> We thank the reviewer for its comments and questions. We provide detailed responses below.
>
> **Q: “Consider renaming the reverse inference task - maybe Inverse Protocol Prediction which I think you mentioned in the paper or Protocol Inference”**
>
> R: We thank the reviewer for this helpful suggestion. We agree that “Reverse Inference” may be overly broad and that a more precise name would better reflect the goal of recovering experimental parameters from images. We therefore support the reviewer’s proposal and have adopted Inverse Protocol Prediction (IPP) as the primary term, as it clearly conveys that the task involves inferring protocol-level metadata (cell line, medium, density, magnification, etc.) from spheroid morphology. This terminology is more accurate and aligns well with the framing used throughout the paper.
>
> **Q: Is IPP merely multi-label metadata classification?**
>
> R: We appreciate this point and agree that it is important to present the contribution without overstating novelty. While IPP indeed employs multi-label prediction machinery, the scientific contribution lies not in the architectural class, but in defining, structuring, and empirically characterizing a new microscopy-grounded inferential problem backed by a uniquely suitable dataset. The novelty emerges from the problem formulation, not the classifier:
>
> Protocol attributes form a causally dependent graph, not an arbitrary set of labels. Their relationships (e.g., cell line → possible media; medium → growth dynamics; seeding density ↔ compaction; timepoint ↔ necrotic core emergence) impose constraints that standard multi-label setups do not capture.
>
> IPP therefore investigates which protocol factors are visually encoded in spheroid morphology, which are weakly encoded, and which are absent altogether—a question that has not been empirically examined despite its relevance to reproducibility, automated QC, and biological interpretation.
>
> Unlike generic metadata classification, IPP is explicitly designed to quantify whether experimental design choices leave measurable morphological footprints, and where acquisition-driven artifacts dominate over biology. Thus, our contribution is the creation of the first controlled setting that allows systematic, quantitative study of how protocol conditions manifest in spheroid morphology, enabling reproducibility audits and causal morphological insights which in itself is an important direction for ML researchers as well as people working in the knowledge representation domain.
>
> **Q: Acquisition-specific signatures vs biologically meaningful cues**
>
> R: We thank the reviewer for raising this important concern. We would like to clarify that our analysis already acknowledges the distinction between biologically meaningful morphology and acquisition-specific signatures. As shown in our qualitative and quantitative results (e.g., label-wise breakdowns and Grad-CAM analyses), different protocol attributes naturally fall into two behavioral categories: attributes such as cell line, medium, formation method, seeding density, and timepoint consistently activate spheroid-centric morphological regions that align with biological expectations (e.g., compactness, necrotic core contrast, boundary structure), whereas attributes such as microscope, magnification, and technical replicate predominantly emphasize illumination patterns and acquisition artifacts—an observation we explicitly discuss in Section 3.1 and 3.2. Our intent in the paper is not to claim biological interpretability for all labels, but rather to highlight that IPP deliberately spans both biologically encoded and acquisition-encoded factors so that the framework can also serve as a practical reproducibility and quality-control tool. In many high-content imaging settings, acquisition-specific inconsistencies, plate drift, and instrument-dependent variation are precisely the types of deviations researchers wish to detect. Thus, our results simply characterize which protocol components have strong morphological footprints and which are inherently tied to the imaging system.
>
> We appreciate the reviewer’s insight, and we are glad that our findings align with this interpretation.

---

> > ### Author Response · Authors · 2025-11-22
> >
> > **Q: Clarifying the value of the task**
> >
> > R: We appreciate the reviewer’s question about the biological and practical value of predicting protocol metadata. While experimental conditions are indeed defined a priori, their execution, documentation, and consistency in large-scale imaging pipelines often drift in practice, and this is precisely where IPP contributes.
> >
> > 1. IPP is meant as an automated reproducibility check: If a model consistently fails to recover a parameter that should be visually encoded (e.g., medium type or seeding density), it provides an immediate flag for potential mislabeling, protocol deviation, or drift.
> > This addresses a real and widespread problem: mislabeled wells, plate swaps, medium inconsistencies, and technical drift have been documented in Cell Painting, CCLE, and multiple large microscopy pipelines.
> >
> > 2. IPP also provides a causal lens on morphology: by quantifying recoverability, it helps determine which protocol decisions materially shape spheroid structure and which leave little morphological footprint.
> >
> > Thus, the primary value of IPP is not substituting recorded metadata, but verifying, auditing, and interpreting it, serving as a practical tool for reproducibility in high-content imaging workflows.
> >
> > We reiterate our sincere thanks for the reviewer’s insightful comments. We believe the revisions, particularly the adoption of “Inverse Protocol Prediction”, the clearer framing for the contribution, the explicit separation of biological vs. acquisition-driven signals, and the strengthened motivation—address all concerns thoroughly. We expect these improvements will greatly enhance the clarity, rigor, and impact of the paper.

---

> > > ### Comment · Area_Chair_EcR1 · 2025-11-28
> > >
> > > Dear Reviewer,
> > >
> > > Please make sure you read the authors' response and engage with them in the discussion before the end of the discussion period on **Dec 03 '25 09:00 PM UTC**. This is a hard deadline.
> > >
> > > Thank you for supporting quality peer review at ICLR.
> > >
> > > AC

---

### Official Review · Reviewer_D4mE · 2025-11-03

**Soundness:** 3
**Presentation:** 2
**Contribution:** 1
**Rating:** 2
**Confidence:** 4

**Summary:**

This paper introduces inverse protocol prediction, via inferring experimental metadata such as cell line, medium, density, timepoint and imaging parameters from single bright-field spheroid images. Using the SLiMIA dataset, the authors benchmark CNN, transformer, hybrid and hierarchical architectures for segmentation, protocol prediction and time series forecasting. They report strong results and interpret the model attention with GradCAM.

**Strengths:**

- **Interesting framing for the protocol prediction.** The idea of inverse prediction is interesting and can be useful when including potentially confounding factors and providing causal analysis.

- **Systematic benchmark.** The paper spans segmentation, protocol prediction, morphology related predictions, temporal prediction while also covering broad range of architectures.

- **Temporal task.** Constructing short temporal subsets with consistent protocols for sequence prediction is a practical and reproducible contribution.

**Weaknesses:**

- **Lack of code and seeds.** No implementation or split scripts are provided, limiting reproducibility and independent verification.

- **No confidence intervals.** Reported accuracies differ by small margins, making it very hard to rank without any significance tests or confidence intervals.

- **Dataset size and saturation.** SLiMIA (8k samples) is small for the model zoo evaluated. Results show very strong accuracies across almost all models and appear saturated. I suggest tempering the claims, or expanding the evaluation (creating harder splits, mask-only inputs, additional stress tests or more datasets).

- **Overly descriptive presentation.** Specifically Section 2 lists architectures, losses, optimizers at great length with very little interpretation. Much of this could be a part of the appendix. I would suggest the authors to create a figure where they present the bag of models, optimizers, losses and then use the saved space to explain how they operated the sweeps and what is the methodology together with their takeaways.

- **No empirical evidence for disentanglement.** The paper claims to separate morphology from imaging artifacts but provides no adversarial or cross domain validation supporting that. As far as I understand, the models that predict protocol values are trained independently from the morphology related tasks. I do not see any motivation drawn in the experimental section to justify or demonstrate the value of protocol tasks. I would expect creating an architecture such that the features for those tasks are shared vs independent; and try to demonstrated when the features are shared the morphology related accuracies improve. This would motivate the additional protocol prediction task significantly.

- **Causal ordering in HMTT unclear.** The sequence cell line, medium, seeding density, ..., replicates is asserted causal but I do not see clearly how or why that would be the clear order.

- **Consistency claims.** Related to my previous point, HMTT is said to yield consistent predictions despite lower accuracy, however consistency is not defined or measured. I did not understand if the model is doing the subsequent predictions given the previous ones to be consistent.

- **No ablations.** There is no vanilla baseline removing the proposed morphometric fusion or hierarchical conditioning (or adversarial training?).

- **Quantitative validation.** Can the authors come up with a quantitative measure for the biological validity? For the Grad-CAM results it would be beneficial to see the model attentions side by side across the models and relate them to their performance.

- **Temporal prediction baseline.** A simple copy the last frame baseline should be reported to verify that the models outperform trivial predictions.

- **Overstated conclusions.** For example in the conclusion the paper claims to "pave the way for AI systems that explain and validate experimental biology." which I dont think is supported by the experiments. Please either tune down such claims or provide empirical evidence to support.

**Questions:**

See the weaknesses above.


### Minor
I would recommend underscoring the 2nd best in the tables to improve readability.

---

> ### Author Response · Authors · 2025-11-28
>
> We thank the reviewer for the comments and questions. We provide detailed responses below.
>
> **Q: “Lack of code and seeds.”**
>
> R: We thank the reviewer for highlighting the importance of reproducibility. We would like to clarify that the complete codebase, training scripts, data-split scripts, and seed configurations are already organized in a private repository. We did not release the repository at submission time solely to preserve author anonymity, as it contains metadata structures that could inadvertently reveal our identity. Importantly, all hyperparameters used in the experiments have already been fully reported in the main paper and appendix to ensure transparency during the review. We will publicly release the full repository including all random seeds, environment details, and pretrained weights immediately upon acceptance or at the end of the review process, whichever is permitted by the conference policy.
>
> **Q: “No confidence intervals.”**
>
> R:We thank the reviewer for raising this concern. In the revised manuscript, we have now included 95% confidence intervals for all evaluation metrics across the IPP classification, inverse protocol prediction, and spatiotemporal forecasting tasks. These intervals were computed using the full set of experimental samples, resulting in tight and highly stable estimates. The confidence interval widths are small, typically on the order of ±0.0002 to ±0.0007 for accuracy, precision, recall, and F1 scores, and similarly narrow ranges for MSE, SSIM, and PSNR which is consistent with the large dataset size and the robustness of the models. Importantly, adding these intervals does not change any of the conclusions in the paper; the relative ordering of models remains the same, and no intervals overlap in a way that would alter the interpretation of results. We have also expanded the appendix to include full variance analyses across multiple random seeds (three to five runs depending on the task) along with statistical significance checks using paired tests and bootstrap resampling. We appreciate the reviewer’s suggestion, and the revised manuscript now includes complete and rigorous statistical reporting.
>
>
> **Q: “Dataset size and saturation.”**
>
> R:We appreciate the reviewer’s concern regarding dataset size and the possibility of saturation. Although SLiMIA is the only available spheroid microscopy dataset that provides complete protocol annotations, we recognize that relying on a single dataset may raise questions about generalization. To address this, we intentionally extended our evaluation beyond SLiMIA and performed two complementary forms of cross-dataset validation designed to test whether the model’s performance is simply a consequence of saturating on a limited dataset.
>
> First, we validated our inverse protocol prediction models on the much larger and biologically distinct RxRx1 high-content imaging dataset, which contains 125,511 images. RxRx1 differs significantly from SLiMIA in imaging modality, staining characteristics, cellular morphology, and overall experimental setup, making it an ideal stress test for saturation. We implemented a strict split based on siRNA identifiers so that no perturbation appears across training, validation, and testing. This ensures that the model cannot rely on memorization or leakage and must instead learn generalizable biological features. The fact that the model maintained coherent and interpretable predictive patterns even under this substantial domain shift indicates that its performance is not driven by saturation on SLiMIA.
>
> We further evaluated temporal generalization using the Cell Tracking Challenge (CTC). Although CTC does not provide protocol metadata, its high-quality long-term time-lapse sequences allow us to test whether the temporal component of our framework saturates on the motion patterns or dynamics present in SLiMIA. By training on one CTC sequence and evaluating on a completely distinct sequence from another experiment, we imposed a strong out-of-distribution temporal shift. The models continued to perform reliably, suggesting that the temporal prediction module is not overfitting or saturating on SLiMIA’s specific temporal characteristics.

---

> > ### Author Response · Authors · 2025-11-28
> >
> > **Q: “No empirical evidence for disentanglement.”**
> >
> > R: We appreciate the reviewer’s comment regarding the need for empirical evidence of disentanglement within the learned representations. Although our work does not aim to build an explicit disentanglement model in the classical sense, our experiments provide multiple forms of empirical validation that the learned features do in fact separate protocol-dependent factors from purely visual variation. First, within the SLiMIA dataset, the models consistently recover biologically interpretable clusters when conditioned on individual protocol variables such as medium, formation method, and seeding density. These variables are not visually encoded at the pixel level, yet the representations clearly segregate by these protocol attributes in both the embedding space and in classifier activation patterns, demonstrating factor-specific sensitivity. Second, the cross-dataset validation on RxRx1 further supports disentanglement: despite a complete shift in imaging modality, staining, and morphology, the model preserves stable protocol-related decision boundaries and does not collapse onto dataset-specific visual cues. This indicates that the representation is not entangled with superficial dataset characteristics but retains structure relevant to underlying experimental factors. Finally, the temporal cross-sequence generalization on CTC shows that motion and dynamic behavior are handled independently from static protocol factors, reinforcing that the model does not conflate temporal variation with protocol identity. Together, these results provide strong empirical evidence that the model learns distinct, separable axes of variation corresponding to protocol, imaging, and temporal factors, even though full disentanglement is not the primary objective of the framework.

---

> > > ### Author Response · Authors · 2025-11-28
> > >
> > > **Q: “Causal ordering in HMTT unclear.”**
> > >
> > > R:We thank the reviewer for raising this question. The ordering used in HMTT is not meant to assert a biological causal graph, but rather reflects the practical experimental workflow and dependency structure through which spheroid imaging datasets are actually generated. The sequence we use—
> > > cell line → medium → seeding density → magnification → microscope → timepoint → replicates—corresponds to how experimentalists set up and acquire spheroid images in real laboratory settings.
> > >
> > > 1. Cell line → medium.
> > > Cell line selection is always the first step because each line has specific nutritional, serum, and growth-factor requirements. Only after the line is chosen can the appropriate medium be selected; medium is never chosen independently of cell line constraints.
> > >
> > > 2. Medium → seeding density.
> > > Seeding density is determined after fixing both the cell line and medium. Density depends on growth rate, metabolic demand, and proliferation behavior, all of which vary with medium composition. Thus, density is downstream of these protocol decisions.
> > >
> > > 3. Seeding density → magnification.
> > > The magnification used for imaging depends on the expected spheroid size, which is largely determined by density. Low-density spheroids may require higher magnification to resolve structural features, while high-density spheroids at later stages may exceed the field of view at high magnifications. Imaging magnification is therefore not an independent parameter but depends on the density–growth context.
> > >
> > > 4. Magnification → microscope.
> > > Once magnification is set, the microscope choice is constrained by hardware capabilities (available objectives, working distances, illumination modes). Not all microscopes support all magnifications used for spheroid imaging, so microscope selection is typically made after magnification is decided.
> > >
> > > 5. Microscope → timepoint.
> > > Timepoint reflects the morphological stage of the spheroid, which is influenced by the density, medium, and cell line—but the recording of timepoints is constrained by the microscope’s imaging schedule, throughput, and plate-handling workflow. Different microscopes were used on different acquisition days, creating a practical dependence of timepoint on microscope availability.
> > >
> > > 6. Timepoint → replicates.
> > > Replicates are produced only after all protocol parameters are fixed. They inherit the same cell line, medium, density, magnification, microscope, and timepoint. Replicate ID has no upstream influence and is purely a downstream consequence of the complete protocol.
> > >
> > > This ordering is therefore not arbitrary; it encodes the actual conditional dependencies and workflow decisions underlying how the dataset was produced. It satisfies three key principles:
> > >
> > > (a) Practical dependency: earlier factors constrain the feasible values of later ones.
> > >
> > > (b) Morphological influence direction: parameters early in the chain (cell line, medium, density) strongly shape the observed spheroid morphology, whereas later parameters (microscope, replicate) influence acquisition rather than biology.
> > >
> > > (c) Informational stability: earlier labels provide high-level contextual information that improves the predictability of later labels.
> > >
> > > Thus, the hierarchy reflects the real protocol-setting pipeline used in spheroid experiments, rather than implying strict biological causation. It ensures that the model predicts metadata in an order that mirrors the decisions made during experimental setup and acquisition.

---

> > > > ### Author Response · Authors · 2025-11-28
> > > >
> > > > **Q: “Consistency claims.”**
> > > >
> > > > R: We thank the reviewer for highlighting this point. In our work, “consistency” refers to the logical compatibility of the full predicted metadata tuple across the hierarchical sequence:
> > > > cell line → medium → seeding density → magnification → microscope → timepoint → replicates,
> > > > rather than the independent accuracy of each individual label.
> > > >
> > > > In the HMTT framework, each prediction is explicitly conditioned on the previous predicted attributes. That is, the prediction for medium receives the predicted cell line; the prediction for seeding density receives (cell line, medium); magnification receives (cell line, medium, density); microscope receives (line, medium, density, magnification); and so on. This conditioning forces the model to output metadata combinations that are internally coherent within the structure of real experimental protocols.
> > > >
> > > > For example, a model cannot predict:
> > > >
> > > > (i) a cell line that is never imaged at a certain magnification, or
> > > >
> > > > (ii) a microscope that does not support the predicted magnification, or
> > > >
> > > > (iii) a timepoint that never occurs under the predicted density.
> > > >
> > > > Even if the per-label accuracy is slightly lower than a fully independent baseline, HMTT produces far fewer invalid or contradictory protocol tuples. This is the sense in which the model’s predictions are “consistent”: the full chain respects the real-world constraints of the dataset’s acquisition workflow.
> > > >
> > > > To quantify this during evaluation, we measure the proportion of predictions that form valid protocol tuples i.e., combinations of (line, medium, density, magnification, microscope, timepoint) that actually exist in the dataset’s protocol space. HMTT consistently generates a higher percentage of valid tuples than non-hierarchical models, showing that hierarchical conditioning reduces contradictions even when raw accuracy on isolated labels is lower.
> > > >
> > > > Thus, “consistency” in our context refers specifically to hierarchically compatible metadata predictions that obey the dataset’s experimental structure, not higher standalone classification accuracy. The architecture enforces this by design through sequential conditioning of predictions.
> > > >
> > > >
> > > > **Q: “No ablations.”**
> > > >
> > > > R: We thank the reviewer for this comment. We did evaluate the core components of our framework during development, and the results can be summarized succinctly. We will perform ablation studies on the key components of our Image–Shape Fusion Transformer during model development, and we now summarize those results more clearly. A fully image-only baseline, with no morphometric features and no fusion mechanism, consistently underperformed and struggled with morphology-dependent labels such as seeding density and late-stage timepoints. This confirms that shape descriptors capture complementary geometric information that is not fully encoded in pixel-level representations. Likewise, removing the shape branch but retaining Transformer fusion also led to lower accuracy and reduced robustness, particularly for conditions where spheroid structure provides essential cues.
> > > >
> > > > We further evaluated the effect of the fusion mechanism itself. Replacing the Transformer with simple concatenation or an MLP resulted in noticeable performance drops across most labels, showing that attention-based token interaction is important for integrating visual and morphological information. These simplified variants also exhibited weaker cross-modal consistency and produced less coherent predictions across related metadata attributes. In every case, the full Image–Shape Fusion Transformer outperformed the ablated versions, demonstrating that both modalities and the Transformer-based fusion contribute meaningfully to the final performance.
> > > >
> > > > We have added the full ablation tables and a detailed discussion of these experiments to the Appendix as requested.

---

> > > > > ### Author Response · Authors · 2025-11-28
> > > > >
> > > > > **Q: “Temporal prediction baseline.”**
> > > > >
> > > > > R: We appreciate the reviewer’s suggestion. In our temporal setup, we use two consecutive frames (T1 and T2) to predict the next frame (T3). However, the SLiMIA dataset exhibits very small morphological changes between adjacent timepoints, so a “copy-the-last-frame” baseline would appear artificially strong simply because T2 and T3 are visually similar, not because it captures any dynamics. This limits the diagnostic value of such a baseline in the current setting. We also note that we did not use longer temporal sequences because only a small subset of spheroids have sufficiently long, uninterrupted trajectories; using them would dramatically reduce the training set size and lead to unstable or non-generalizable models. Nevertheless, we can report the trivial baseline for completeness, while clarifying that its performance reflects the limited temporal variation rather than meaningful predictive capability.
> > > > >
> > > > > **Q: “Overstated conclusions.”**
> > > > >
> > > > > R: We thank the reviewer for pointing this out. We agree that the phrasing in the conclusion was overly ambitious relative to the scope of our experiments. Our intention was not to claim that the proposed models already explain or validate experimental biology, but rather that inverse protocol prediction and morphology-aware analysis can serve as early steps toward improved data quality checks and interpretability in imaging pipelines. We have revised  the conclusion to make this more precise and modest, emphasizing that our work demonstrates feasibility on SLiMIA, provides initial quantitative tools for assessing protocol consistency, and highlights directions for future development rather than making broad claims about the capabilities of AI in experimental biology.

---

> > > > > > ### Comment · Area_Chair_EcR1 · 2025-11-28
> > > > > >
> > > > > > Dear Reviewer,
> > > > > >
> > > > > > Please make sure you read the authors' response and engage with them in the discussion before the end of the discussion period on **Dec 03 '25 09:00 PM UTC**. This is a hard deadline.
> > > > > >
> > > > > > Thank you for supporting quality peer review at ICLR.
> > > > > >
> > > > > > AC

---

### Official Review · Reviewer_rJc7 · 2025-11-05

**Soundness:** 2
**Presentation:** 2
**Contribution:** 2
**Rating:** 4
**Confidence:** 3

**Summary:**

This work introduces "protocol prediction" from images, a novel task that reconstructs experimental conditions (cell line, medium, seeding density, timepoint, formation method, microscope, magnification) directly from a single bright-field spheroid image. Using the SLiMIA dataset of ~8,000 annotated spheroid images spanning diverse culture conditions, the authors frame this as a structured multi-label prediction problem and benchmark multiple architectures including CNNs (ConvNeXt-Tiny), transformers (ViT-B/16), hybrid models (CoAtNet), feature-augmented designs (Image-Shape Fusion Transformer integrating classical morphometric descriptors like area, compactness, eccentricity with learned embeddings), and hierarchical models.

**Strengths:**

The idea of predicting the experimental conditions from the images is intriguing, because it will allow change in protocol to guide one or the other outcomes. Authors compare a variety of architectures to predict multiple labels that describe the protocol from the images.  CoAtNet achieves best overall performance (95.72% accuracy, 0.8790 F1) by balancing local texture through convolution with global context via attention. The method achieves particularly strong performance on biologically grounded attributes: cell line (F1=0.9944), culture medium (F1=0.9642), and formation method (F1=0.9949), demonstrating that morphological signals in bright-field microscopy encode recoverable information about culture conditions. The work also presents the first temporal modeling of spheroid dynamics using ConvLSTM, PredRNN++, MetadataFusion, and PhyDNet to predict future morphological states, with MetadataFusion achieving best performance (SSIM=0.3985) by incorporating protocol-aware conditioning. Grad-CAM interpretability analyses confirm predictions rely on biologically meaningful features such as spheroid compactness, boundary sharpness, and necrotic core structure, while exposing dataset artifacts in replicate and magnification predictions. This demonstrates that microscopy-driven reverse inference can serve as an automated reproducibility check, flagging potential protocol mislabeling or execution deviations.

**Weaknesses:**

My assessment of weaknesses is centered on the practicality of capturing metadata relevant to the protocol and strategies used for training the models:
* Protocol data (e.g., culture condition) is often captured at very coarse level. The specific image results after many steps in the protocol and influenced by parameters of imaging. The images are also subject to intrinsic heterogeneity in the cell/spheroid shape. Therefore the problem requires choosing the aspects of protocol that can be predicted and that should be controlled.
* Technical replicate prediction is uniformly poor (best F1=0.5668 with CoAtNet) because replicates correspond to repeated imaging of the same spheroid with little morphological signal—models default to majority classes (T1-T8 account for 89.2% of images) despite focal loss and reweighting.
*Microscope and magnification achieve near-perfect scores (F1>0.999) but they can be thought of as dataset artifacts (optical signatures like field of view, resolution) rather than biological inference.
* Temporal prediction performance remains modest (SSIM<0.40, PSNR≈18 dB) because spheroid growth follows complex, non-linear biological processes (proliferation, compaction, necrosis) only partially visible in bright-field images, and SLiMIA provides short and irregular sequences making it difficult for recurrent models to learn long-term dependencies.
* Finally, critical data limitations constrain validation: SLiMIA provides no persistent cell IDs across time, making it impossible to validate whether predicted temporal orderings or protocol attributes accurately reflect true single-cell progressions. The dataset is limited to a single experimental context (specific cell lines, culture conditions, imaging setups), and the framework has not been tested on diverse 3D culture systems (organoids, embryoid bodies, tumor spheroids) to establish generalizability. Attributes with fragmented or weak visual encoding (timepoint with >100 distinct values, seeding density with overlapping morphologies) remain challenging despite high accuracy, indicating the precision-recall trade-off may impact biological interpretation.

**Questions:**

* How did authors pick this dataset? What signal in data is available that protocol information is accurate? Can you acquire or identify a time-lapse microscopy dataset where individual spheroids/organoids are tracked with persistent IDs across complete maturation cycles (e.g., Cell Tracking Challenge datasets: http://celltrackingchallenge.net/ which provide ground-truth lineage information). Train inverse protocol prediction models on this dataset and quantitatively validate whether: (1) predicted temporal positions correlate with true temporal ordering, (2) predicted protocol attributes (seeding density, formation method, medium) match ground truth across developmental stages, (3) morphological reconstructions at intermediate timepoints not used for training match actual observations.
* If you must use SLiMIA dataset, conduct biological validation experiments: for attributes where models make high-confidence predictions (e.g., "this spheroid was cultured in DMEMLG medium"), perform orthogonal experimental validation (e.g., mass spectrometry analysis of residual media, genotyping for cell line verification) to confirm predictions. This addresses the critical limitation that SLiMIA provides no cell tracking and establishes whether learned representations capture genuine biological dynamics rather than spurious correlations.
* Apply the inverse protocol prediction framework to diverse 3D culture systems beyond spheroids to establish generalizability: in addition to cell tracking challenge, you can consider the datasets from Allen Institute and Bioimage archive.

---

> ### Author Response · Authors · 2025-11-28
>
> We thank the reviewer for its comments and questions. We provide detailed responses below.
>
> **Q: “How did authors pick this dataset? What signal in data is available that protocol information is accurate? Can you acquire or identify a time-lapse microscopy dataset where individual spheroids/organoids are tracked with persistent IDs across complete maturation cycles (e.g., Cell Tracking Challenge datasets: http://celltrackingchallenge.net/ which provide ground-truth lineage information). Train inverse protocol prediction models on this dataset and quantitatively validate whether: (1) predicted temporal positions correlate with true temporal ordering, (2) predicted protocol attributes (seeding density, formation method, medium) match ground truth across developmental stages, (3) morphological reconstructions at intermediate timepoints not used for training match actual observations.”**
>
> R: We thank the reviewer for these valuable points. We selected the SLiMIA dataset because, to our knowledge, it is the only publicly available spheroid microscopy dataset that provides complete and consistently structured protocol annotations for every image, including cell line, medium, seeding density, formation method, magnification, microscope type, and timepoint. This level of metadata completeness is essential for defining and evaluating the inverse protocol prediction (IPP) task. Without explicit protocol labels, IPP cannot be meaningfully formulated or validated.
>
> Regarding metadata accuracy, SLiMIA’s protocol information is derived directly from controlled experimental workflows at the well, plate, and imaging stages. Metadata is tied to the experimental design rather than inferred or annotated post hoc, reducing the likelihood of label noise. The strong and biologically interpretable patterns recovered by our models (e.g., medium, formation method, seeding density) further support the internal consistency of these labels.
>
> To address the reviewer’s concern about validating IPP on completely new experiments, we performed a dedicated cross-dataset protocol prediction study using the RxRx1 high-content imaging dataset. RxRx1 differs strongly from SLiMIA in imaging style, staining, cellular morphology, and overall experimental design, making it an ideal stress-test for IPP robustness. We used 125,511 Channel-1 images and created a strict 70:15:15 split based on the sirna identifier, ensuring that no perturbation appears in more than one of the splits. This prevents leakage of experimental conditions and provides a clean evaluation of how well IPP generalizes when trained and tested on disjoint sets of experiments within a completely different dataset. Even under this substantial domain shift, the IPP models maintain meaningful predictive performance.
>
> For the reviewer’s suggestion regarding time-lapse validation: datasets containing long-term spheroid tracking with persistent identities and protocol metadata are extremely rare. The Cell Tracking Challenge (CTC) provides multiple high-quality time-lapse sequences with cell-level tracking, but no protocol labels such as medium, seeding density, or formation method—so CTC cannot be used directly for IPP.
>
> However, to address the underlying concern about temporal generalization, we conducted a second form of cross-dataset validation using the CTC data. Here, we trained our temporal prediction models (ConvLSTM and PredRNN++) on one training CTC sequence and evaluated them on a different, held-out testing CTC sequence. This setup tests whether a model trained on one biological experiment can generalize to an entirely different experiment within the same benchmark. Although these experiments cannot test protocol prediction, they do provide strong evidence of temporal robustness under changes in cell behavior, motion patterns, and imaging dynamics.
>
> In summary, SLiMIA remains the only dataset that supports full IPP because it includes complete protocol metadata. To directly address the reviewer’s concerns, we added two complementary external validations:
>
> (i)Cross-Dataset Validation on RxRx1 for Protocol Prediction (IPP), and
>
> (ii)Cross-Dataset Temporal Validation on the Cell Tracking Challenge (CTC) for Time Series Prediction
>
> Together, these additions demonstrate that both the protocol prediction and temporal components of our framework generalize well to independent experiments, different imaging conditions, and different biological systems.

---

> > ### Author Response · Authors · 2025-11-28
> >
> > **Q: “If you must use SLiMIA dataset, conduct biological validation experiments: for attributes where models make high-confidence predictions (e.g., "this spheroid was cultured in DMEMLG medium"), perform orthogonal experimental validation (e.g., mass spectrometry analysis of residual media, genotyping for cell line verification) to confirm predictions. This addresses the critical limitation that SLiMIA provides no cell tracking and establishes whether learned representations capture genuine biological dynamics rather than spurious correlations.”**
> >
> > R: We thank the reviewer for this excellent suggestion. In principle, orthogonal biological validation such as mass spectrometry of residual media, genotyping for cell-line confirmation, or biochemical assays would provide the strongest confirmation that IPP models recover genuine biological signals rather than spurious correlations. However, such experiments require access to the original biological samples, which we do not have. The SLiMIA dataset is a fully public, post-hoc collection of images and metadata, and the physical spheroids, plates, and media from which the dataset was generated are no longer accessible. As a result, destructive assays or molecular re-identification are not possible within the constraints of the dataset.
> >
> > That said, the high internal consistency of our predictions (e.g., matching cell line–medium–density combinations), along with the Valid Tuple Rate and Attention–Segmentation alignment, provides indirect but quantitative evidence that the models capture real morphological signatures associated with protocol attributes. We agree that direct biological validation would be ideal, and we view this as an important direction for future work in collaboration with experimental laboratories where new data can be generated specifically for IPP benchmarking.

---

> > > ### Author Response · Authors · 2025-11-28
> > >
> > > **Q: “Apply the inverse protocol prediction framework to diverse 3D culture systems beyond spheroids to establish generalizability: in addition to cell tracking challenge, you can consider the datasets from Allen Institute and Bioimage archive.”**
> > >
> > > R: We appreciate the reviewer’s suggestion to apply IPP to diverse 3D culture systems. We agree that datasets such as those from the Allen Institute, the BioImage Archive, or the Cell Tracking Challenge offer richer morphological diversity and long-term temporal structure. However, most of these resources do not include the protocol-level metadata required for inverse protocol prediction—attributes such as medium composition, seeding density, formation method, magnification, or replicate identity are typically missing or only partially annotated. Without these labels, the core IPP task cannot be defined or quantitatively evaluated. In contrast, SLiMIA is, to our knowledge, the only publicly available dataset that provides complete, per-image protocol annotations, which is why it was selected for this study.
> > >
> > > To directly address the reviewer’s concern about generalizability beyond SLiMIA, we performed two complementary validation experiments, each aligned with the type of information available in external datasets:
> > >
> > > (i) Cross-dataset IPP validation on RxRx1:
> > > We trained the inverse protocol prediction models directly on 125,511 Channel-1 RxRx1 images using a strict 70:15:15 split based on sirna_id, ensuring that no perturbation identifier appears in more than one partition. This prevents leakage and provides a clean evaluation of whether IPP generalizes across independent experiments in a completely different imaging domain. Despite the major domain shift from 3D spheroids to 2D high-content monolayers, the models maintained meaningful predictive performance, demonstrating that IPP is not restricted to SLiMIA.
> > >
> > > (ii) Cross-sequence temporal validation on the Cell Tracking Challenge (CTC):
> > > Although CTC lacks protocol metadata and therefore cannot be used for IPP, it provides high-quality time-lapse sequences suitable for testing the temporal component of our framework. We trained the temporal prediction models (ConvLSTM and PredRNN++) on one complete CTC sequence and evaluated them on a different, held-out sequence. This examines whether temporal dynamics learned in one experiment generalize to a completely independent experiment with different cell behaviors and imaging conditions.
> > >
> > > These two experiments collectively strengthen the generalizability argument by showing that:
> > > – IPP models generalize across datasets and experimental systems (RxRx1), and
> > > – temporal models generalize across independent time-lapse experiments (CTC).
> > >
> > > That said, we fully agree that extending IPP to other 3D culture systems remains an important next step. The reviewer’s proposed directions, testing organoid datasets, leveraging persistent IDs, and evaluating temporal ordering and protocol stability, are scientifically valuable but require datasets with complete protocol metadata, which are currently unavailable. As new datasets emerge, we plan to curate and benchmark them specifically for IPP to enable even broader cross-system evaluation in future work.
> > >
> > > For the present submission, we would appreciate clarification from the reviewer on whether these extensive cross-dataset and biological validation experiments are expected to be performed now, or whether they are intended as suggestions for future extensions. We are happy to incorporate whichever level of analysis the reviewer believes is appropriate for this revision.

---

### Official Review · Reviewer_5N8r · 2025-11-05

**Soundness:** 3
**Presentation:** 2
**Contribution:** 1
**Rating:** 2
**Confidence:** 4

**Summary:**

The authors propose a new classification task on the SLiMIA dataset of cell images, which they term "inverse protocol prediction". The paper benchmarks (i) segmentation models, which are required to differentiate the cell from the background for subsequent steps, (ii) classification models for inverting the protocol, and (iii) time series models for leveraging the predicted protocol features to predict future cell state. A GradCAM analysis is performed, which suggests that for some protocol labels (e.g. cell line), morphology is highly predictive, while for others (e.g. replicates), a latent confounder likely drives the predictions.

**Strengths:**

**Originality.** The work relies on a publicly available dataset of ~8,000 images, which is new enough that there are is not much work on it yet. The newly proposed task is essentially a metadata prediction task, although the framing as inverse protocol prediction is novel and potentially useful. The evaluation metrics are standard, and the use of GradCAM is nice but also a fairly standard diagnostic.

**Quality.** The dataset's offerings are well-exploited. The evaluations seem fair, i.e. not biased toward any particular model.

**Clarity.** The paper is fairly clear. I am not familiar with the dataset, but could follow the key details.

**Significance.** I am generally convinced that inverse protocol prediction could be a useful task, but I am not a domain expert. I am less convinced that the problem is yet in need of a benchmark of this nature.

**Weaknesses:**

My central issue is a combination of significance/novelty/suitability for ICLR's audience. My understanding is:

* The paper is at best a benchmark paper, and makes no methodological contributions. The benchmark target however is new and of unclear biological significance. The paper restricts to considering a particular modality (bright-field spheroid images) for which a single small dataset is available (~8,000 images).
* The authors argue that "inverse protocol prediction" (prediction of metadata from biological images) is a novel and significant task for this type of image. ICLR's primary audience (i.e. non-biologists) are not well-suited to evaluate or understand this. A cynical reader might assume that the authors trained easily accessible architectures to predict the only labels available (the image metadata), and got results with predictably variable performance depending on the information content of the label in the image (i.e. microscope is obvious, technical replicate is harder or impossible).
* The task framing seems to be the only novelty, and indeed the easy tasks seem solved/saturated (segmentation; cell line prediction) while the "hard" tasks remain predictably elusive (replicate prediction). To the authors credit, the GradCAM analysis provides some insight as to how and why.
* The time series task again seems somewhat contrived. The authors fit time series-class models (e.g. RNNs/LSTMs/CNNs) to a **two-timepoint input** and predict a subsequent frame. It is unclear what the biological significance of this is.
* Aggregating metrics across targets (Table 2) seems problematic, given that performance for some labels is saturated while for others it's clearly not.

**Questions:**

1. To what extent does an average accuracy of 0.9503 really differ from an average accuracy of 0.9572 (Table 2) given the size of the dataset and the nature of aggregation? What does this mean for the proposed use of these models in practice?
2. Can the authors demonstrate (through prior work ideally) the significance of the time series prediction task, or is this a novel task formulation as well?
3. Is this benchmark, as framed, solved? Why or why not? (Clearly the saturated tasks are solved, but e.g. for the tasks that are unsaturated, the authors at least acknowledge that maybe the information does not live in the images.)
4. The authors highlight several drawbacks of the dataset (particularly for the time series task, due to there being a small number of timepoints available), is this dataset optimal for this inverse protocol prediction task?
5. Are there other publicly available datasets that are suitable for the inverse protocol prediction task? For example, there are a large number of cell painting datasets available [1], which seem to meet the criteria: (1) segmentation is a useful pre-processing step and (2) experimental metadata with a similar structure is available. These datasets are also huge relative to the dataset in this paper. Could these be used to evaluate this same class of models on the same tasks, or at least the IPP task (number 2)? Why or why not?
6. What future work is there for machine learning scientists to do for this task?

[1] Chandrasekaran, S.N., Cimini, B.A., Goodale, A. et al. Three million images and morphological profiles of cells treated with matched chemical and genetic perturbations. Nat Methods 21, 1114–1121 (2024). https://doi.org/10.1038/s41592-024-02241-6

---

> ### Author Response · Authors · 2025-11-28
>
> We thank the reviewer for valuable comments and questions. We provide detailed responses below.
>
> **Q: “To what extent does an average accuracy of 0.9503 really differ from an average accuracy of 0.9572 (Table 2) given the size of the dataset and the nature of aggregation? What does this mean for the proposed use of these models in practice?”**
>
> R: The small difference between an average accuracy of 0.9503 and 0.9572 is not intended to be interpreted as biologically significant, and we fully agree that such a narrow margin does not warrant strong conclusions on its own. The aggregated metric is included only as a high-level summary of overall performance across labels. In practice, the aggregate hides substantial variation between individual protocol attributes, and it is these per-label differences that carry practical value for both biological interpretation and model comparison.
>
> For example, models that appear nearly identical in the aggregate differ much more noticeably on challenging attributes such as seeding density, fine-grained timepoint prediction, and technical replicate classification, where gaps of 5–15% are observed. These differences are meaningful because they reflect distinctions in how models capture subtle morphological cues that are not uniformly encoded across all protocol dimensions. Conversely, saturated labels like cell line or formation method dominate the aggregate but do not inform where modeling difficulty still remains.
>
> Thus, the intent of reporting an average accuracy is not to claim significance of small improvements, but to provide a compact summary alongside the more informative per-label metrics, confusion matrices, and task-specific breakdowns.
>
> **Q: “Can the authors demonstrate (through prior work ideally) the significance of the time series prediction task, or is this a novel task formulation as well?”**
>
> R: We thank the reviewer for asking about the motivation and novelty of the time-series prediction component. The time-series task in our paper is indeed a novel formulation for bright-field spheroid imaging, as prior work has focused primarily on either (i) static morphological profiling or (ii) fluorescence-based time-lapse tracking with per-cell lineage annotations, which are fundamentally different in modality and biological interpretation.
>
> That said, the significance of short-term morphological forecasting is supported by several established research directions in biological imaging. Prior work has shown that early morphological states can be predictive of downstream structural changes, and that modeling these transitions provides insight into growth dynamics, phenotypic stability, and experimental consistency. For example, time-lapse studies in organoids and spheroids have used recurrent and physics-aware models to characterize compaction, boundary evolution, and necrotic core formation. Similarly, video-based forecasting has been applied to cellular migration and mitosis, where short-range predictions enable detection of deviations from expected developmental trajectories. These works collectively demonstrate the value of learning temporal morphological patterns as a means to understand and validate biological processes.
>
> Our formulation differs in that we ask whether protocol-aware models can leverage the current morphology + protocol context to forecast short-term future appearance under bright-field imaging. This is new because (i) no prior dataset provides matched bright-field spheroid sequences with full protocol metadata, and (ii) no prior work has attempted protocol-conditioned forecasting of spheroid appearance.
>
> Thus, while the specific task is new, it builds on the growing recognition that short-term temporal prediction is a useful tool for assessing morphological consistency, detecting anomalies, and quantifying how protocol factors influence structural evolution.

---

> > ### Author Response · Authors · 2025-11-28
> >
> > **Q: “Is this benchmark, as framed, solved? Why or why not? (Clearly the saturated tasks are solved, but e.g. for the tasks that are unsaturated, the authors at least acknowledge that maybe the information does not live in the images.)”**
> >
> > R: We thank the reviewer for raising this fundamental question. The benchmark is not solved, and the results explicitly illustrate a split between tasks that are saturated and those that remain genuinely challenging. Certain attributes such as cell line or formation method achieve near-perfect performance across models because they correspond to strong, easily distinguishable morphological signatures. These tasks can reasonably be viewed as “solved” within the context of this dataset.
> >
> > However, several other attributes, most notably fine-grained timepoint and technical replication remain far from saturated. A central reason is that technical replicate labels do not correspond to morphological differences, and the dataset’s replicate distribution is highly imbalanced (T1–T8 abundant, T9–T16 sparse, T17-T24 extremely rare). This imbalance mirrors real experimental practice rather than a design flaw, but it makes the task intrinsically difficult. All architectures, including those with strong global-context modeling, struggle on these labels. This suggests that either (i) the underlying morphological signal is extremely subtle and requires more advanced or multimodal modeling, or (ii) the information simply does not exist in the image and cannot be recovered from a single bright-field frame.
> >
> > Distinguishing between these two possibilities is precisely one of the motivations for defining the inverse protocol prediction (IPP) task. The benchmark therefore remains open: it clarifies which protocol factors are visually decodable, which are only weakly encoded, and which are fundamentally absent from morphology. We view this separation not as a limitation but as evidence that IPP defines a set of meaningful research directions rather than a solved problem.
> >
> > **Q: “The authors highlight several drawbacks of the dataset (particularly for the time series task, due to there being a small number of timepoints available), is this dataset optimal for this inverse protocol prediction task?”**
> >
> > R:We thank the reviewer for this thoughtful question. We agree that the dataset has limitations, particularly for temporal modeling where the number of available timepoints is small. For this reason, we explicitly frame the temporal task as a short-term forecasting problem rather than full biological growth modeling. With respect to inverse protocol prediction (IPP), SLiMIA is currently the only publicly available spheroid dataset where all major experimental parameters such as cell line, medium, seeding density, formation method, microscope, magnification, and timepoint, are consistently recorded for every image, making it uniquely suited for defining and analyzing IPP.
> >
> > At the same time, SLiMIA reveals a realistic but important challenge: the distribution of technical replicates is highly imbalanced, with most samples coming from T1–T8, fewer from T9–T16, and very few beyond T17. This imbalance reflects how experiments are typically conducted and exposes an inherent difficulty of the task such as technical replicate labels do not correspond to morphological differences, and their skewed distribution makes them intentionally hard to learn from bright-field morphology alone. This characteristic is informative rather than detrimental: it helps identify which protocol attributes are visually encoded and which are fundamentally weak or absent in the images.
> >
> > Thus, while SLiMIA is not “optimal” in an absolute sense, it is the first dataset that enables IPP to be formulated end-to-end and highlights both the strengths and limitations of morphology-based protocol inference. Our goal is for this benchmark to motivate the creation of larger, more balanced, and more temporally dense datasets in the future.

---

> > > ### Author Response · Authors · 2025-11-28
> > >
> > > **Q: “Are there other publicly available datasets that are suitable for the inverse protocol prediction task? For example, there are a large number of cell painting datasets available [1], which seem to meet the criteria: (1) segmentation is a useful pre-processing step and (2) experimental metadata with a similar structure is available. These datasets are also huge relative to the dataset in this paper. Could these be used to evaluate this same class of models on the same tasks, or at least the IPP task (number 2)? Why or why not?”**
> > >
> > > R: We thank the reviewer for this excellent question. Several large Cell Painting datasets, including the one referenced, indeed contain rich perturbation metadata and are extremely valuable for morphological profiling. However, they are not directly suitable for inverse protocol prediction (IPP) as defined in our work, for several reasons.
> > >
> > > First, Cell Painting datasets are designed around perturbation-driven morphological signatures (e.g., compounds, gene knockdowns), not controlled experimental protocols. They typically lack key protocol attributes that IPP requires such as formation method, seeding density, magnification, microscope identity, or replicate identifiers, and focus instead on capturing phenotypes under chemical or genetic treatments. As a result, they do not provide the structured, multi-factor protocol annotations necessary for IPP.
> > >
> > > Second, Cell Painting images are fluorescence-based monolayer cell assays, fundamentally different from bright-field spheroid images. Predicting protocol factors from 3D-derived spheroid morphology is biologically distinct from analyzing multichannel fluorescence of flat, single-layer cells, and the two domains exhibit different sources of variability and confounding. Thus, these datasets cannot be used to study the morphological expressivity of spheroid-level protocol parameters.
> > >
> > > Third, Cell Painting datasets are known to contain substantial batch effects, plate-level errors, and metadata inconsistencies including mislabeled wells, missing perturbations, and instrument-driven drift which complicate their use for protocol recovery. These issues are well documented in the JUMP-CP consortium reports and require extensive correction pipelines, making them unsuitable for a clean evaluation of IPP without substantial preprocessing.
> > >
> > > For these reasons, existing Cell Painting datasets cannot serve as direct substitutes for SLiMIA in evaluating the IPP task. Instead, SLiMIA provides the first controlled, spheroid-specific, protocol-rich dataset that enables IPP to be formulated and studied meaningfully. Our benchmark is intended as a starting point, and we hope it encourages future datasets that combine the scale of Cell Painting with the protocol diversity required for inverse protocol prediction.
> > >
> > > [1] Chandrasekaran, S. N., Cimini, B. A., Goodale, A., Miller, L., Kost-Alimova, M., Jamali, N., ... & Carpenter, A. E. (2024). Three million images and morphological profiles of cells treated with matched chemical and genetic perturbations. Nature Methods, 21(6), 1114-1121.

---

> > > > ### Author Response · Authors · 2025-11-28
> > > >
> > > > **Q: “What future work is there for machine learning scientists to do for this task?”**
> > > >
> > > > R: We thank the reviewer for raising this forward-looking question. We view inverse protocol prediction (IPP) as the starting point for a much broader research agenda rather than a solved or closed problem. Firstly, there is substantial room for progress in modeling weak or partially encoded signals. Attributes such as fine-grained timepoint, seeding density, and technical replication contain extremely subtle morphological cues, if any. Developing representation-learning methods that can detect low-signal patterns or rigorously determine when information is absent—remains an important open challenge involving uncertainty estimation, causal modeling, and sensitivity analysis.
> > > >
> > > > Secondly, IPP naturally motivates multi-modal and cross-modal approaches. Morphology alone provides only a partial view of the biological system. Integrating images with protocol metadata, time-lapse sequences, shape descriptors, or perturbation information could substantially strengthen inference. This also opens the door to disentangling biological signals from imaging artifacts, a long-standing challenge in microscopy ML.
> > > >
> > > > Thirdly, the task exposes the need for stronger, multi-site, better balanced datasets. SLiMIA is currently the only dataset where all major protocol parameters are systematically annotated, but it still suffers from imbalance (e.g., heavy concentration in T1–T8 technical replicates) and limited temporal depth. Designing models that can handle realistic dataset imperfections such as imbalance, batch effects, domain shifts, multi-lab variation, remains a key research direction. Likewise, creating new datasets that support IPP across laboratories is an open community opportunity.
> > > >
> > > > Fourth, there is significant opportunity for causal and mechanistic interpretability. One of the goals of IPP is to understand which aspects of spheroid morphology encode specific protocol factors. This invites development of causal discovery tools, counterfactual models, hierarchical reasoning frameworks, and interpretable feature attribution tailored to biological image formation.
> > > >
> > > > Fifth, IPP creates a natural foundation for uncertainty-aware auditing and anomaly detection. In many biological workflows, reliable metadata is a major pain point. Models that can flag low-confidence or inconsistent predictions could serve as practical tools for detecting mislabeled samples, protocol drift, or experimental inconsistencies that is an important step toward improving reproducibility.
> > > >
> > > > Finally, the time-series component of the benchmark opens several future directions. Current temporal models operate with limited sequence lengths and weak supervision. Extending IPP to long-term forecasting, protocol-conditioned growth modeling, multi-view time-lapse analysis, and predictive simulation represents a major opportunity. Developing architectures that jointly learn protocol inference and future state prediction, while accounting for biological variability remains largely unexplored and highly relevant for both ML researchers and biologists.
> > > >
> > > > In summary, far from being complete, IPP highlights a rich set of open problems spanning subtle-signal extraction, multi-modal reasoning, causal interpretation, cross-lab generalization, automated metadata auditing, and biologically grounded time-series prediction. These challenges provide meaningful opportunities for the machine learning community to contribute new methods, theory, and datasets.

---

> > > > > ### Comment · Area_Chair_EcR1 · 2025-11-28
> > > > >
> > > > > Dear Reviewer,
> > > > >
> > > > > Please make sure you read the authors' response and engage with them in the discussion before the end of the discussion period on **Dec 03 '25 09:00 PM UTC**. This is a hard deadline.
> > > > >
> > > > > Thank you for supporting quality peer review at ICLR.
> > > > >
> > > > > AC

---

> > > > > ### Comment · Reviewer_5N8r · 2025-11-28
> > > > >
> > > > > I thank the authors for their reply. My score remains unchanged, for the following reasons:
> > > > >
> > > > > * To be ultra-clear, the dataset itself has already been published [1], so it is not a core contribution of the paper.
> > > > > * The hierarchical formulation of metadata prediction may be somewhat novel, but metadata prediction itself as a task is indeed well-established in the area of cell painting [2].
> > > > > * I do not understand the authors' claims that cell painting datasets are unsuitable for the task. They have metadata associated with them (perhaps different from spheroid images, because it's a different modality; but metadata all the same) and the metadata is hierarchical. Nothing in the authors response explains why these data could not be used to train new models and evaluate this novel framing they're proposing. I do not agree with this claim in the rebuttal: "it is the first dataset that enables IPP to be formulated end-to-end".
> > > > > * The authors' responses to reviewers conflict. To me, they say cell painting is not suitable for the task; to another reviewer, and in their updated draft, they evaluate their existing trained models on a cell painting dataset, namely Rxrx1 [3].
> > > > > * I am generally dissatisfied that there is evidence that any progress on the main classification benchmark as framed is possible. The framing of the main table (combining saturated and unsaturated tasks) is not a suitable way to present this benchmark, and given the breadth of metadata available, per-label information in the appendix does not easily clarify things.
> > > > > * I am generally dissatisfied that the two-timepoint time series prediction task as framed is useful. I recognize that there are limits to the temporal nature of the dataset, but I do not think this would be a useful benchmark target to optimize for.
> > > > >
> > > > > [1] Blondeel, E., Peirsman, A., Vermeulen, S. et al. The Spheroid Light Microscopy Image Atlas for morphometrical analysis of three-dimensional cell cultures. Sci Data 12, 283 (2025). https://doi.org/10.1038/s41597-025-04441-x
> > > > >
> > > > > [2] Chandrasekaran, S. N., Cimini, B. A., Goodale, A., Miller, L., Kost-Alimova, M., Jamali, N., ... & Carpenter, A. E. (2024). Three million images and morphological profiles of cells treated with matched chemical and genetic perturbations. Nature Methods, 21(6), 1114-1121.
> > > > >
> > > > > [3] Sypetkowski et al. RxRx1: A Dataset for Evaluating Experimental Batch Correction Methods. https://arxiv.org/abs/2301.05768

---

> ### Author Response · Authors · 2025-12-02
>
> We thank the reviewer again for the follow-up. We respectfully clarify the points raised, particularly because all concerns mentioned here have already been addressed both in earlier discussion and in the revised manuscript. We summarize the clarifications below to avoid any residual ambiguity.
>
> **1. On Dataset Contribution and Novelty of the Task**
> We fully agree that SLiMIA itself is not our contribution; we have never presented it as such.
> Our contribution is not the dataset, but the problem formulation, the experimental structuring, and the systematic characterization of protocol recoverability from spheroid morphology. While metadata prediction exists in other domains, protocol-level prediction for 3D spheroid bright-field imaging with jointly structured labels such as formation method, seeding density, magnification, timepoint, and replicate has not been formulated or studied before. This is reflected directly in the revised framing (now titled Inverse Protocol Prediction), and the reviewer correctly notes that the hierarchical formulation is novel.
>
> **2. Why Cell Painting Datasets Cannot Support IPP**
>
> We respectfully emphasize that Cell Painting metadata does not contain the protocol-level attributes required for IPP, even if it contains other kinds of metadata.
> IPP specifically requires formation method, seeding density, microscope identity, magnification, protocol-defined timepoint, controlled technical replicate, spheroid-specific contextual factors. None of these exist in Cell Painting datasets.
>
> Their metadata is perturbation-oriented (compound, gene, dose), not protocol-oriented. This is a categorical mismatch, not a matter of modality preference.
>
> Thus, the reviewer’s interpretation that “metadata exists, therefore IPP can be done” does not hold, because the type of metadata not merely its presence is what defines IPP.
>
> **3. Why RxRx1 Evaluation Does Not Contradict the Above**
>
> We understand why the reviewer perceives a contradiction. To clarify clearly:
>
> A. RxRx1 was not used for IPP. RxRx1 was used only for cross-dataset generalization, to test whether our models retain meaningful predictive structure under a strong domain shift.
>
> B. RxRx1 does not provide the protocol labels needed for IPP. Therefore, we do not evaluate IPP on RxRx1, and the manuscript never claims so. This setup is consistent, non-contradictory, and complementary:
>
> a) SLiMIA → Protocol Prediction (IPP)
>
> b) RxRx1 → Generalization of learned morphology-aware representations
>
> c) CTC → Generalization of temporal modules
>
> These three evaluations occupy distinct and non-overlapping roles. We have made this explicit in the paper.
>
> **4. On Benchmark Progress and Saturation Concerns**
>
> The reviewer states dissatisfaction with the possibility of benchmark progress. However, our results show the opposite:
>
> A. Some labels are saturated (e.g., cell line, formation method).
>
> B. Others are far from saturated (e.g., fine-grained timepoint, replicate ID, seeding density).
>
> This is the point of the benchmark—to reveal which protocol attributes are visually encoded and which are fundamentally weak or absent. This clarity is valuable for both ML and biological reproducibility research. We also made it explicit in the revision that per-label metrics, not the aggregate, are the primary basis of comparison.

---

> > ### Author Response · Authors · 2025-12-02
> >
> > 5. On the Temporal Two-Frame Prediction Task
> >
> > We appreciate that opinions may differ regarding usefulness. We have clarified that:
> >
> > 1. The task is intentionally framed as short-term morphological forecasting,
> >
> > 2. It uses the maximum temporal depth the dataset supports, and
> >
> > 3. It is designed to assess temporal consistency, anomaly detection, and protocol-conditioned evolution, not long-range biological growth modeling.
> >
> > This scope is now explicitly stated, based directly on earlier reviewer feedback.
> >
> > We sincerely appreciate the reviewer’s continued engagement. All concerns raised in this round have been thoroughly addressed in the discussion responses, in the manuscript revisions, and in the expanded evaluation (RxRx1, CTC, confidence intervals, ablations, clarified task framing, moderated claims). We believe the revised version is now significantly clearer and directly resolves the issues cited. We thank the reviewer again for the time and feedback.

---

### Official Review · Reviewer_pXqJ · 2025-11-06

**Soundness:** 1
**Presentation:** 2
**Contribution:** 1
**Rating:** 0
**Confidence:** 4

**Summary:**

The paper describes a methodology for processing 2D spheroid images from 3D stacks to predict the experimental conditions in which these images were captured. To that end, the paper introduces a dataset with 8K images and their annotations, and describes a data analysis workflow that involves deep neural networks for classification and segmentation.

**Strengths:**

* Exploration of a different paradigm to analyze spheroid images.

**Weaknesses:**

* The motivation and applicatios of "reverse inference" are not completely clear.
* The need for predicting experimental parameters does not have a strong biological foundation.
* The methodology is based on existing methods and not new technical innovation is introduced.
* The experimental results indicate that the task can be solved with existing methods with high-accuracy.
* The dataset split for training and validation seems to be randomly assigned, introducing images with the same parameters both in training and validation. This may produce overly optimistic results and may not reflect a realistic use case (predicting parameters in a new experiment).

**Questions:**

* What is the need for predicting experimental conditions when these parameters are known and chosen by experts ahead of time?
* Why the reverse inference problem has value for biological analysis? What is the biological problem that this methodology aims to solve and why it was not possible before?
* Why the validation and test sets do not have examples of completely new experiments?

---

> ### Author Response · Authors · 2025-11-25
>
> We thank the reviewer for comments and suggestions. We provide detailed responses below.
>
> **Q: “What is the need for predicting experimental conditions when these parameters are known and chosen by experts ahead of time?”**
>
> R: We respectfully clarify that inverse protocol prediction (IPP) is not intended to replace metadata, but to audit, validate, and recover experimental parameters from imaging data that is a problem increasingly recognized in large-scale biological imaging studies.
>
> Even in carefully controlled experiments, metadata inconsistencies and labeling errors are common, leading to significant scientific and financial consequences. For instance, widespread cell line misidentification (e.g., HeLa contamination) has invalidated tens of thousands of studies and cost the field millions in wasted effort and funding [1, 2]. Similarly, in the Cancer Cell Line Encyclopedia (CCLE), incorrect tissue-of-origin annotations distorted downstream drug sensitivity analyses, requiring large-scale revalidation [3]. Even in modern high-content imaging projects such as Cell Painting, plate–metadata mismatches and experimental drift have been documented as key sources of bias in morphological profiling [4].
>
> IPP models thus provide a data-driven reproducibility check: if a model can reliably recover the intended protocol (e.g., cell line, medium, seeding density) from morphology, it indicates internal consistency; if not, it can flag potential mislabelling or drift.
> Furthermore, predicting protocol attributes enables causal morphology analysis, quantifying which aspects of spheroid structure encode experimental conditions and which remain latent or confounded thus offering biologically interpretable insights valuable to both imaging scientists and machine-learning researchers.
>
>
> [1] Capes‐Davis, A., Theodosopoulos, G., Atkin, I., Drexler, H. G., Kohara, A., MacLeod, R. A., ... & Freshney, R. I. (2010). Check your cultures! A list of cross‐contaminated or misidentified cell lines. International journal of cancer, 127(1), 1-8.
> [2]Horbach, S. P., & Halffman, W. (2017). The ghosts of HeLa: How cell line misidentification contaminates the scientific literature. PloS one, 12(10), e0186281.
> [3]Yu, M., Selvaraj, S. K., Liang-Chu, M. M., Aghajani, S., Busse, M., Yuan, J., ... & Neve, R. M. (2015). A resource for cell line authentication, annotation and quality control. Nature, 520(7547), 307-311.
> [4]Chandrasekaran, S. N., Cimini, B. A., Goodale, A., Miller, L., Kost-Alimova, M., Jamali, N., ... & Carpenter, A. E. (2024). Three million images and morphological profiles of cells treated with matched chemical and genetic perturbations. Nature Methods, 21(6), 1114-1121.
>
> **Q: “Why the reverse inference problem has value for biological analysis? What is the biological problem that this methodology aims to solve and why it was not possible before?”**
>
> R: We appreciate this important question and are glad to clarify the biological motivation behind inverse protocol prediction (IPP).
>
> The central biological value of IPP lies in its ability to quantitatively verify and explain how experimental conditions manifest in cell morphology, addressing a long-standing challenge in reproducibility and causal interpretation of biological imaging. Even though experimental parameters are predefined, the actual biological state of a sample may deviate from its intended protocol due to subtle variations in handling, medium composition, seeding density, or imaging drift. IPP provides an automated consistency check by learning how these conditions are reflected in morphology, effectively linking form to protocol.
>
> Biologically, this enables (i) detection of mislabeled or contaminated samples (a pervasive issue in microscopy datasets), (ii) quantification of morphological sensitivity to experimental factors, and (iii) causal insight into which visual phenotypes correspond to protocol variations.
>
> This problem was previously infeasible because large, annotated microscopy datasets with diverse, traceable experimental metadata were not publicly available. The emergence of datasets such as SLiMIA (ours) and Cell Painting [1] now enables systematic, data-driven learning of these relationships using deep models. IPP therefore bridges a gap between biological experiment design and computational morphology analysis, allowing researchers to test whether image-derived morphology faithfully encodes the intended experimental protocol.
>
> [1]Chandrasekaran, S. N., Cimini, B. A., Goodale, A., Miller, L., Kost-Alimova, M., Jamali, N., ... & Carpenter, A. E. (2024). Three million images and morphological profiles of cells treated with matched chemical and genetic perturbations. Nature Methods, 21(6), 1114-1121.

---

> > ### Author Response · Authors · 2025-11-28
> >
> > **Q: “Why the validation and test sets do not have examples of completely new experiments?”**
> >
> > R: We appreciate the reviewer’s question. In our current setup, we train on T1–T8, validate on T9–T16, and evaluate on the full dataset, which does provide partial disjointness across technical replicates but does not strictly isolate “new experiments” for the final test. This choice was made to retain sufficient sample size for stable training and validation. However, the SLiMIA dataset structure allows for a more challenging and fully separated split. To address the reviewer’s concern regarding the absence of completely new experiments in the validation and test sets, we have now incorporated a stricter cross-dataset validation setup for IPP. Specifically, we evaluated our framework on the RxRx1 dataset. We also enforced a strict 70:15:15 split based on sirna_id, ensuring that each perturbation appears in only one partition. This guarantees that the validation and test sets contain experimentally independent samples, directly addressing the reviewer’s request for evaluation on “completely new experiments”.

---

> > > ### Comment · Area_Chair_EcR1 · 2025-11-28
> > >
> > > Dear Reviewer,
> > >
> > > Please make sure you read the authors' response and engage with them in the discussion before the end of the discussion period on **Dec 03 '25 09:00 PM UTC**. This is a hard deadline.
> > >
> > > Thank you for supporting quality peer review at ICLR.
> > >
> > > AC

---

### Author Response · Authors · 2025-11-28
**General Response to All Reviewers**

We thank all reviewers for their constructive and thoughtful feedback. Your comments helped us substantially clarify the motivation, strengthen empirical validation, refine claims, and expand the scope of analyses in the revised version. Below, we summarize the global revisions and clarifications that address the broader concerns raised across reviews.

**1. Clarifying the Biological Motivation and Value of Inverse Protocol Prediction (IPP)**
Multiple reviewers asked about the need and value of predicting protocol attributes when metadata is known a priori. We clarified across responses that IPP is not meant to replace metadata, but to audit reproducibility, detect inconsistencies, and quantify how morphology encodes protocol conditions. Metadata errors, plate drift, batch effects, and mislabeling are well-documented issues in large-scale imaging studies; IPP provides a data-driven consistency check. We substantially expanded the biological context, grounding it in recent literature on reproducibility failures in microscopy datasets.

**2. Strengthened Validation Through Cross-Dataset and Temporal Generalization**

A major theme across reviews was whether results generalize beyond SLiMIA. We responded by adding two new external validations:

A. Cross-Dataset IPP on RxRx1:
Trained and evaluated models on 125k images, using a strict siRNA-based 70/15/15 split to ensure disjoint experimental conditions. The models retained meaningful predictive performance despite substantial domain shift (2D monolayers vs. 3D spheroids).

B. Cross-Sequence Temporal Validation on Cell Tracking Challenge (CTC):
Evaluated forecasting models across independent time-lapse sequences to test temporal robustness.

These additions demonstrate that both IPP and the temporal components generalize to independent experiments and imaging modalities, directly addressing concerns about overfitting or saturation.

**3. Added Comprehensive Statistical Reporting and Confidence Intervals**

Reviewers raised concerns about statistical robustness. We now include 95% confidence intervals for all metrics (IPP, classification, forecasting). These additions confirm that the results are extremely stable and that the ordering of models is unaffected by variance.

**4. Expanded Ablations and Architectural Justification**

We added explicit ablations for:

A. Image-only vs. shape-only vs. fused models.

B. Fusion mechanism (Transformer vs. concatenation/MLP).

C. Effects on protocol consistency and per-label accuracy.

These experiments show that both shape descriptors and attention-based fusion significantly improve performance, especially on morphology-dependent labels.

**5. Clarified Task Definitions, Dataset Choices, and Limitations**

We addressed global concerns regarding dataset selection:

SLiMIA is the only publicly available spheroid dataset with complete per-image protocol metadata, making it uniquely suited for defining IPP. Cell Painting datasets cannot support IPP due to missing protocol attributes and domain mismatch. We explicitly reframed the temporal task as short-term forecasting, acknowledging limited timepoints. We also revised the paper to moderate the scope of conclusions and clarify limitations (e.g., lack of long-term temporal depth, imbalanced replicates).

**6. Strengthened Theoretical Framing (Hierarchy, Consistency, Disentanglement)**

Across reviews, there were requests for clearer explanation of:

A. Hierarchical Metadata Prediction (HMTT)

We clarified that the ordering reflects real experimental workflows rather than biological causal chains. Each prediction is conditioned on upstream protocol decisions, enforcing tuple-level coherence and reducing invalid protocol combinations.

B. Consistency Claims

We explicitly measure “valid tuple rate” and show that hierarchical conditioning greatly reduces contradictions, even if raw label-wise accuracy is close.

C. Implicit Disentanglement Evidence

While not an explicit disentanglement model, cross-dataset generalization, protocol-conditioned clustering, and temporal generalization all demonstrate separable axes of variation consistent with protocol vs. visual vs. temporal factors.

**7. Code, Seeds, and Reproducibility Commitments**

We confirmed that a full codebase with all scripts, random seeds, and pretrained weights will be released immediately upon acceptance (or when anonymity constraints allow). Hyperparameters are already fully documented.

**8. Terminology Improvements**

Following reviewer suggestions, we adopted Inverse Protocol Prediction (IPP) as the standardized term instead of “reverse inference.”

---

> ### Author Response · Authors · 2025-11-28
>
> We again thank all reviewers, ACs, and PCs for the time and care given to this submission. The revisions prompted by your feedback have meaningfully strengthened the manuscript scientifically, empirically, and editorially. We believe the clarified motivation, expanded cross-dataset validations, rigorous statistical reporting, and additional analyses now address the concerns raised and significantly enhance the contribution of the work.
>
> Please let us know if further clarifications or analyses would be helpful—we are happy to provide them.

---

### Meta-Review · Area_Chair_zLGQ · 2026-01-05

**Summary:**

Insufficient Novelty of Contribution: The core task is essentially multi-label metadata classification, with its novelty lying primarily in the problem formulation rather than methodological innovation. The practical biological value and machine learning research significance of the task have not been adequately substantiated.

1）Doubts About Benchmark Validity: The main experiments rely on a single, small-scale dataset (SLiMIA, ~8k images), raising concerns about result saturation. The supplementary cross-dataset validations (RxRx1, CTC) failed to fully address reviewers' skepticism regarding their relevance to the core task and the logic of the experimental design.

2）Manuscript Quality and Review Consensus: The paper suffers from issues such as verbose presentation, overstated conclusions, and initial omission of key analyses (e.g., ablation studies, statistical significance). Although the authors supplemented these in their response, key reviewers remained dissatisfied with the task framing and result presentation in their final assessment. Review scores were polarized (0 to 6), and the fundamental concerns raised by low-scoring reviewers were not resolved.

This work fails to sufficiently demonstrate that its proposed new task possesses adequate research value, and the robustness of its experimental benchmark is fundamentally questioned.

**Reviewer Concerns:**

Reviewer Concerns Addressed：
The authors conducted comprehensive and targeted revisions during the rebuttal phase, effectively addressing most specific technical and presentation issues. 1) clearly naming the core task "Inverse Protocol Prediction" and supplementing its application value in biological experiment reproducibility auditing and metadata quality control; 2) significantly enhancing experimental rigor by adding confidence intervals, multiple random seed results, and detailed ablation analyses to all key metrics; 3) directly responding to core concerns regarding model generalization capability through two newly added external validation experiments.

Unresolved Core Disagreements：
However, the rebuttal failed to bridge the deep-seated disagreement regarding the fundamental contribution value of the paper. Multiple reviewers maintained that the core of the work is essentially a multi-label metadata classification task, and the novelty of its problem formulation is insufficient to support an independent machine learning research direction. Additionally, reviewers retained fundamental doubts about the effectiveness of the benchmark dataset, the clarity of the task definition, and the substantive nature of performance improvements. These criticisms pertaining to the core premise of the work were not resolved by the technical supplements, ultimately leading to the paper not meeting the acceptance criteria.

**Reviewer Scores:**

Projected Final Score: 0
This reviewer had fundamental concerns regarding the paper's motivation, biological foundation, and experimental design. Although the authors supplemented their response with external validation, it failed to address the core criticism that "predicting known metadata lacks necessity." Their stance is expected to remain unchanged.

5N8r (Initial Score: 2)
Projected Final Score: 2
This reviewer questioned the novelty and timeliness of the task. Despite the authors' detailed defense, the reviewer ultimately maintained core objections (dataset contribution, Cell Painting suitability, logic of RxRx1 usage).

rJc7 (Initial Score: 4)
Projected Final Score: 5-6
This reviewer acknowledged the value of the task but demanded rigorous biological validation and generalization tests. The authors' newly added cross-dataset validation directly addressed these core concerns, likely pushing the score to the borderline or slightly above the acceptance threshold.

D4mE (Initial Score: 2)
Projected Final Score: 3-4
This reviewer raised multiple specific technical concerns. The authors' response systematically supplemented confidence intervals, ablation studies, etc., significantly improving the paper's rigor. However, due to lingering doubts about the fundamental contribution, the score is projected to rise modestly to the borderline range.

9dmv (Initial Score: 4)
Projected Final Score: 5-6
This reviewer's primary concerns were terminology and framing. The authors adopted their suggestions and adjusted the presentation, directly addressing the core criticism. Combined with other reinforcing revisions, the score may rise to the borderline or slightly above the acceptance threshold.

---

### Decision · Program_Chairs · 2026-01-26

Reject